# Morpho-Molecular Characterization of Hypocrealean Fungi Isolated from Rice in Northern Thailand

**DOI:** 10.3390/jof11040321

**Published:** 2025-04-18

**Authors:** Sahar Absalan, Alireza Armand, Ruvishika S. Jayawardena, Nakarin Suwannarach, Jutamart Monkai, Nootjarin Jungkhun Gomes de Farias, Saisamorn Lumyong, Kevin D. Hyde

**Affiliations:** 1Department of Biology, Faculty of Science, Chiang Mai University, Chiang Mai 50200, Thailand; saharabsalan6@gmail.com (S.A.); suwan.462@gmail.com (N.S.); mjutamart@gmail.com (J.M.); 2Center of Excellence in Microbial Diversity and Sustainable Utilization, Faculty of Science, Chiang Mai University, Chiang Mai 50200, Thailand; 3Center of Excellence in Fungal Research, Mae Fah Luang University, Chiang Rai 57100, Thailand; armandalireza.72@gmail.com (A.A.); ruvishika.jay@mfu.ac.th (R.S.J.); 4School of Science, Mae Fah Luang University, Chiang Rai 57100, Thailand; 5Office of Research Administration, Chiang Mai University, Chiang Mai 50200, Thailand; 6Department of Plant Pathology, Faculty of Agriculture at Kamphaeng Saen, Kasetsart University Kaphaeng Saen Campus, Nakhon Pathom 73140, Thailand; nootjarin.nan@gmail.com; 7Rice Department, Chiang Rai Rice Research Center, Phan, Chiang Rai 57120, Thailand; 8Academy of Science, The Royal Society of Thailand, Bangkok 10300, Thailand; 9Department of Plant Pathology, College of Agriculture, Guizhou University, Guiyang 550025, China

**Keywords:** *Fusarium*, hypocreales, phylogeny, taxonomy, saprobe

## Abstract

Hypocreales is one of the largest orders within the class Sordariomycetes and is renowned for its diversity of lifestyles, encompassing plant, insect, and human pathogens, as well as endophytes, parasites, and saprobes. In this study, we focused on saprobic hypocrealean fungi isolated from rice in northern Thailand. Species identification was conducted using morphological characteristics and multilocus phylogenetic analyses, including the internal transcribed spacer region (ITS), 28S large subunit nuclear ribosomal DNA (LSU), translation elongation factor 1–alpha (*tef*1-α), RNA polymerase II second-largest subunit (*rpb*2), and calmodulin (*cmdA*). This research confirmed the presence of 14 species of hypocrealean taxa, viz. *Fusarium* (9), *Ochronectria* (1), *Sarocladium* (2), *Trichothecium* (1), and *Waltergamsia* (1). Among these were two new species (*Fusarium chiangraiense* and *F. oryzigenum*), four new host records (*Fusarium kotabaruense*, *Ochronectria thailandica*, *Sarocladium bactrocephalum*, and *Waltergamsia fusidioides*), and three new geographical records (*Fusarium commune*, *F. guilinense*, and *F. hainanese*).

## 1. Introduction

*Oryza sativa* L., commonly known as rice, is a crucial cereal crop and a vital source of fiber, minerals, proteins, and vitamins, promoting good health for more than half of the world’s population [1,2]. In Thailand, rice is a major economic crop cultivated in all regions of the country, making Thailand one of the leading producers and exporters of rice [3,4,5]. However, many factors, especially microorganisms such as fungi, can significantly impact rice supply [6]. Rice-associated fungi, which exhibit diverse nutritional modes, play key roles in rice production, ranging from causing severe diseases to providing beneficial effects [7,8,9].

Hypocreales ranks among the largest orders within the class Sordariomycetes, comprising 358 genera across 29 families, with some genera incertae sedis [10,11,12,13]. The taxonomy of families in this order has been fairly well studied [14,15,16,17,18,19,20,21,22,23,24]. They have worldwide distribution across a wide range of hosts and habitats [25]. Hypocreales are considered the richest source of biocontrol agents within the Ascomycota, containing species with significant ecological and economic roles [25,26,27,28,29,30,31]. They are also found as crop pathogens, saprobes, entomopathogens, and mycoparasites [23,25,32,33,34,35,36,37,38]. In general, families within this order exhibit distinct associations with specific hosts. For instance, *Nectriaceae* encompasses many important plant pathogens, including *Fusarium*; *Cordycipitaceae* consists of well-known insect pathogens and medicinal fungi; and *Clavicipitaceae* includes grass endophytes that provide benefits to plants [16,39,40,41]. The association between hypocrealean fungi and the family Poaceae, particularly rice, has been documented in many studies. Certain genera, including *Acremonium*, *Fusarium*, and *Sarocladium*, have been most frequently observed and isolated from rice [42,43,44,45,46,47,48,49,50,51,52,53,54,55,56]. The present study aims to investigate saprobic hypocrealean taxa associated with rice, with an emphasis on *Fusarium* in northern Thailand.

## 2. Materials and Methods

### 2.1. Sample Collection, Isolation, and Examination

Rice specimens, predominantly consisting of the Thai Jasmine rice cultivars RD15 and KDML105, were collected mainly during the dry season (November and December) from paddy fields in different districts of Chiang Rai Province, Thailand. Samples were placed in plastic bags with labels detailing their collection information and transferred to the laboratory. External observations and examinations were conducted using a stereomicroscope (Motic SMZ 168), and microscopic images were captured using a digital camera attached to a compound microscope (Nikon DS-Ri2). The images featured in the figures were edited with Adobe Photoshop version 21.1.3 (Adobe Systems, San Jose, CA, USA) and measured using the Tarosoft^®^ Image Framework (version 0.9.7).

The fungal isolation and purification of obtained strains were carried out using the hyphal tip technique as described by Senanayake et al. [57]. Some of the isolates were transferred onto carnation leaf agar (CLA) plates and incubated at 27 °C for 7–14 days to induce sporulation [58]. Morphological characteristics were assessed either directly from the host or after a two-week incubation period of the obtained cultures. Specimens were deposited at Mae Fah Luang University Herbarium (Herb. MFLU) and living cultures on PDA were preserved in the Culture Collection of Mae Fah Luang University (MFLUCC). New taxa were assigned numbers from Index Fungorum (https://www.indexfungorum.org/names/names.asp, accessed on 25 November 2024).

### 2.2. DNA Extraction, PCR Amplification, and Sequencing

Fresh mycelia from axenic cultures were scraped for DNA extraction using the OMEGA E.Z.N.A.^®^ Forensic DNA Kit (Norcross, GA, USA) according to the manufacturer’s instructions. Polymerase chain reaction (PCR) amplifications were performed for internal transcribed spacer region (ITS), 28S large subunit nuclear ribosomal DNA (LSU), translation elongation factor 1–alpha (*tef*1-α), RNA polymerase II second largest subunit (*rpb*2), and calmodulin (*cmdA*) using related forward and reverse primer pairs [59,60,61,62,63]. The PCR thermal cycle program and primers used in this study are listed in Table 1. The total volume of PCR mixtures was 25 µL, which contained double-distilled water (ddH_2_O), 12.5 μL of 2 × Power Taq PCR MasterMix (*Taq* DNA polymerase, dNTPs, MgCl_2_, and optimized reaction buffers), 1 μL each of the primers (10 pM), and 1 μL of genomic DNA. All of the PCR products were assessed using 1.5% agarose gel electrophoresis and viewed on a Cybergreen-stained agarose gel using ultraviolet light with a gel imaging system. DNA fragments with positive results in the PCR analysis were subsequently shipped to SolGent Co. (Daejon, Republic of Korea) for purification and sequencing.

### 2.3. Phylogenetic Analyses

Phylogenetic analyses followed the procedures described by Dissanayake et al. [64]. Sequence data for all loci were subjected to a BLASTn search (Basic Local Alignment Search Tool, https://blast.ncbi.nlm.nih.gov/Blast.cgi, accessed on 17 July 2024) to identify the most closely related taxa from the NCBI (National Center for Biotechnology Information) database. Each selected sequence was aligned using the server version of MAFFT v.7.036 [65], and the alignments were manually edited using BioEdit v.7.0.5.2 before conducting phylogenetic analyses [66]. The FASTA files of combined alignment were converted to PHYLIP and NEXUS formats using the Alignment Transformation Environment (ALTER) online tool (https://www.sing-group.org/ALTER, accessed on 17 July 2024) [67] for Maximum Likelihood (ML) and Bayesian analysis, respectively. The ML trees were constructed using IQ-TREE on XSEDE v.2.3.2 [68] via the CIPRES Science Gateway platform [69] implementing the ultrafast bootstrap with 1000 replicates. Bayesian posterior probability (PP) was calculated using MrBayes version 3.1.2 [70] with Markov Chain Monte Carlo sampling (MCMC). Four simultaneous Markov chains were run for 1,000,000 generations, with sampling every 100th generation. The first 25% of trees from the burn-in phase were discarded, while the remaining 75% were used to compute the posterior probability (PP). Phylogenetic trees were visualized using FigTree v.1.4.0 [71] and edited with Adobe Illustrator CC 22.0.0 (Adobe Systems, San Jose, CA, USA).

## 3. Results

### 3.1. Phylogenetic Analysis

Analysis 1: A multilocus phylogenetic analysis of the *Fusarium fujikuroi* species complex (FFSC) was carried out using *cmdA*, *rpb*2, and *tef*1-α gene sequences, comprising 99 ingroup taxa. *Fusarium curvatum* (CBS 238.94) and *F. inflexum* (CBS 716.74) were used as outgroups. The aligned dataset included 1855 positions: *cmdA* (1–800), *rpb*2 (801–1330), and *tef*1-α (1331–1855), with gaps considered. The best ML tree yielded a final likelihood score of −11755.683 (Figure 1). The alignment showed 683 distinct patterns, 418 parsimony-informative sites, 171 singleton sites, and 1295 constant sites. The Bayesian tree converged at the 1,000,000th generation with an average standard deviation of split frequencies of 0.195795. This phylogeny resolved four regional lineages: African clades A and B, and American and Asian clades. Strains MFLUCC 24-0637 and MFLUCC 24-0628 were placed within African clade A and the Asian clade, respectively.

Analysis 2: A phylogenetic reconstruction of the *Fusarium nisikadoi* species complex (FNSC) was conducted using a combined dataset of *rpb*2 and *tef*1-α from 13 ingroup isolates, with *F. udum* (BBA 65058) as the outgroup. The alignment spanned 1467 characters: *rpb*2 (1–901) and *tef*1-α (902–1330), with gaps included. The optimal IQ-TREE yielded a final likelihood score of −2874.566 (Figure 2). The matrix comprised 118 distinct patterns, 60 parsimony-informative and singleton sites each, alongside 1343 constant sites. The Bayesian tree converged at the 1,000,000th generation with an average standard deviation of split frequencies of 0.001498.

Analysis 3: The combined phylogenetic analysis of *cmdA*, *rpb*2, and *tef*1-α conducted on *Fusarium incarnatum*-*equiseti* species complex (FIESC) isolates included 101 ingroup taxa, with *Fusarium concolor* (CBS 961.87) as the outgroup. The dataset after alignment consisted of 2153 characters: 1–648 for *cmdA*, 649–1525 for *rpb*2, and 1526–2153 for *tef*1-α, including gaps. The best scoring IQ-TREE, with a value of −12,330.550 as a final ML optimization likelihood, is presented in Figure 3. The matrix comprised 796 distinct patterns, 490 parsimony-informative sites, 287 singleton sites, and 1376 constant sites. The Bayesian tree converged at the 1,000,000th generation with an average standard deviation of split frequencies of 0.038027.

Analysis 4: Phylogenetic analysis of *Ochronectria* isolates was performed using a combined dataset from ITS, LSU, *rpb*2, and *tef*1-α gene regions, involving six ingroup taxa. *Lasionectria marigotensis* (CBS 131606) was designated as the outgroup. The sequence alignment totaled 2808 characters: 1–493 for ITS, 494–1270 for LSU, 1271–2005 for rpb2, and 2006–2808 for tef1-α, including gaps. The maximum likelihood tree yielded a final score of −5572.845 (Figure 4). This alignment comprised 144 distinct patterns, 104 parsimony-informative sites, 228 singleton sites, and 2476 constant sites. The Bayesian tree converged at the 1,000,000th generation with an average standard deviation of split frequencies of 0.002432.

Analysis 5: For *Sarocladium* species, a multi-gene phylogeny was conducted using ITS, LSU, and act loci, with 54 ingroup taxa. *Chlamydocillium curvulum* (CBS 430.66 and CBS 229.75) were used as outgroups. The aligned dataset consisted of 1944 characters: 1–425 for ITS, 426–1202 for LSU, and 1203–1944 for *act*, including gaps. The ML analysis returned a best tree with a likelihood score of −9784.272 (Figure 5). The matrix comprised 491 distinct patterns, 375 parsimony-informative sites, 65 singleton sites, and 1504 constant sites. The Bayesian tree converged at the 1,000,000th generation with an average standard deviation of split frequencies of 0.007275.

Analysis 6: The combined phylogenetic analysis of ITS, LSU, and *tef*1-α conducted on *Trichothecium* isolates included 14 ingroup taxa, with *Stanjemonium spectabile* (CBS 340.70) as an outgroup. The dataset after alignment consisted of 2077 characters: 1–491 for ITS, 492–1269 for LSU, and 1270–2077 for *tef*1-α, including gaps. The best scoring IQ-TREE, with a value of −4281.827 as a final ML optimization likelihood, is presented in Figure 6. The matrix had 144 distinct patterns, 164 parsimony-informative sites, 91 singleton sites, and 1873 constant sites. The Bayesian tree converged at the 1,000,000th generation with an average standard deviation of split frequencies of 0.002635.

Analysis 7: Phylogenetic relationships of *Waltergamsia* species were inferred using a combined ITS, LSU, *rpb*2, and *tef*1-α dataset with 26 ingroup taxa. Outgroup taxa included *Acremonium egyptiacum* (CBS 114785) and *A. gamsianum* (CBS 881.73). The alignment comprised 2840 characters: 1–501 for ITS, 502–1278 for LSU, 1279–2032 for *rpb*2, and 2033–2840 for *tef*1-α, including gaps. The best scoring IQ-TREE, with a value of −13,746.021 as a final ML optimization likelihood, is presented in Figure 7. The matrix comprised 720 distinct patterns, 660 parsimony-informative sites, 169 singleton sites, and 2011 constant sites. The Bayesian tree converged at the 1,000,000th generation with an average standard deviation of split frequencies of 0.002520.

### 3.2. Taxonomy

***Fusarium chiangraiense*** S. Absalan, S. Lumyong & K. D. Hyde, sp. nov. (Figure 8).

Index Fungorum Number: IF903755

Etymology: Named after Chiang Rai Province, from where it was collected.

*Saprobic* on *Oryza sativa*. *Conidiophores* 30–95 μm tall, borne on aerial mycelia, septate, proliferating percurrently, bearing terminal or lateral phialides, sympodial and branched, smooth- and thin-walled, pink to hyaline. *Conidiogenous cells* 9.5–17 × 2.5–3.5 μm ($\bar{x}$ = 12 × 3 μm, *n* = 20), mono- and polyphialidic, subulate to subcylindrical, smooth and thin-walled, hyaline. *Microconidia* 6–9 × 2.5–3 μm ($\bar{x}$ = 7 × 3 μm, *n* = 30), single on the tips of phialides, ovoid to ellipsoidal, aseptate, smooth- and thin-walled, hyaline. *Macroconidia* 37–52 × 3.5–4.5 μm ($\bar{x}$ = 46 × 4 μm, *n* = 30), straight to falcate, slender and sometimes slightly curved, tapering toward the basal part, apical cells papillate, basal cells indistinct or foot-shaped, 3–4-septate, hyaline. *Sporodochia* and *Chlamydospores* not observed.

Culture characteristics: Colonies on PDA reaching 67–70 mm diameter after a week at 28 °C, pale greyish rose, cottony to velvety, raised, aerial mycelia medium-dense, filiform margin. Reverse pale orange with pale red at the center.

Material examined: Thailand, Chiang Rai Province, Phan District, Mueang Phan Sub-district (19.484667° N, 99.720012° E), on the panicle of *Oryza sativa*, 9 November 2021, Nootjarin Jungkhun, (NS27-1 = MFLU 25-0033); (ex-type, living culture MFLUCC 24-0628).

GenBank numbers: *cmdA* = PV297810, *rpb*2 = PV394842, *tef*1-α = PV394838.

Notes: Based on the morphological and molecular data, *Fusarium chiangraiense* has been confirmed as a new species, supported by 99% ML and 1.00 PP (Figure 1). The phylogram indicates that *F. chiangraiense* is closely related to *F. globosum* in the FFSC; however, it diverges by 17 base pairs across the combined three-locus dataset (*cmdA*+*rpb*2+*tef*1-α). *Fusarium globosum* is characterized by two types of microconidia: globose and ellipsoidal or clavate, with the latter sometimes forming chains or false heads. It also typically produces macroconidia on sporodochia [72,73]. In contrast, *F. chiangraiense* lacks both globose microconidia and sporodochia. It can also be differentiated by the red mycelium that it forms on carnation leaves.

***Fusarium commune*** K. Skovg., O’Donnell & Nirenberg, *Mycologia* 95: 632 (2003). (Figure 9).

Index Fungorum Number: IF489435

*Saprobic* on *Oryza sativa*. *Conidiophores* 20–50 μm tall or reduced to conidiogenous cells borne terminally or laterally on aerial mycelia, unbranched, smooth- and thin-walled, hyaline. *Conidiogenous cells* 5.5–30 × 2.5–3.5 μm ($\bar{x}$ = 22 × 3.5 μm, *n* = 20), monophialidic (polyphialidic conidiogenous cells were not observed), subulate to subcylindrical, smooth- and thin-walled, hyaline. *Aerial conidia* 6.5–65 × 2.5–4.5 μm ($\bar{x}$ = 33 × 4 μm, *n* = 40), cylindrical, straight to slightly curved, smooth- and thin-walled, 0–3-septate, microcyclic conidiogenesis observed, hyaline. *Sporodochia* not observed. *Chlamydospores* 7.5–10 mm diameter, intercalary or terminal, smooth, hyaline.

Culture characteristics: Colonies on PDA reaching 45–53 mm diameter after a week at 28 °C, pink to pale violet, dense, floccose to fluffy, slightly undulate margin. Reverse coral pink with white margin.

Material examined: Thailand, Chiang Rai Province, Phan District, Mueang Phan Sub-district (19.528477° N, 99.74594° E), on the flag leaf of *Oryza sativa*, 3 December 2021, Nootjarin Jungkhun, (NS40-1 = MFLU 25-0036); (living culture MFLUCC 24-0631).

GenBank numbers: *cmdA* = N/A, *rpb*2 = PV394843, *tef*1-α = PV394839.

Notes: Phylogenetic analysis revealed that our strain clustered with the ex-type strain (CBS 110090) and other *Fusarium commune* isolates within the FNSC, with 98% ML and 1.00 PP support (Figure 2). Combining morphological characteristics and molecular data, strain MFLUCC 24-0631 was confirmed as *F. commune*. While most morphological features were similar between the ex-type strain and our collection, we did not observe polyphialidic conidiogenous cells in our sample.

***Fusarium guilinense*** M.M. Wang, Qian Chen & L. Cai, Persoonia 43: 80 (2019). (Figure 10).

Index Fungorum Number: IF829535

*Saprobic* on *Oryza sativa*. *Conidiophores* 46–75 × 3.5–5 ($\bar{x}$ = 60 × 4.5 μm, *n* = 8), borne on aerial mycelia, unbranched, bearing terminal or lateral phialides. *Conidiogenous cells* mono- and polyphialidic, slightly cylindrical, smooth- and thin-walled, hyaline. *Aerial conidia* 12–58 × 4–6 μm ($\bar{x}$ = 45 × 5.5 μm, *n* = 30), ellipsoid to falcate, slender, smooth- and thin-walled, slightly curved, tapering at both apex and base, 1–5-septate, microcyclic conidiogenesis observed, hyaline. *Sporodochia* and *Chlamydospores* not observed.

Culture characteristics: Colonies on PDA reaching 46–48 mm diameter after a week at 28 °C, white, velvety to felty, irregular margin, aerial mycelium scant in the center. Reverse white to pale salmon.

Material examined: Thailand, Chiang Rai Province, Mueang Chiang Rai District, on the panicle of *Oryza sativa*, 10 November 2021, Ruvishika Jayawardena, (RS14-1 = MFLU 25-0032); (living culture MFLUCC 24-0626).

GenBank numbers: *cmdA* = PV297811, *rpb*2 = PV394827, *tef*1-α = PV394837.

Notes: Our isolate grouped with *Fusarium guilinense* (CGMCC 3.19495) in the phylogenetic analysis, with 100% ML and 1.00 PP support (Figure 3). Both isolates are morphologically similar, however, our isolate predominantly produced 3–5-septate conidia, whereas Wang et al. [74] described conidia in their strain as mostly 3-septate. In the present study, we documented *F. guilinense* as the first geographical record for Thailand.

***Fusarium hainanense*** M.M. Wang, Qian Chen & L. Cai, *Persoonia* 43: 82 (2019). (Figure 11).

Index Fungorum Number: IF829536

*Saprobic* on *Oryza sativa*. *Conidiophores* borne on aerial mycelia, septate, bearing terminal or lateral phialides, smooth- and thin-walled, hyaline. *Conidiogenous cells* 11–22 × 2.5–3.5 μm ($\bar{x}$ = 18 × 3 μm, *n* = 15), mono- and polyphialidic, subulate to subcylindrical, smooth- and thin-walled, hyaline. *Aerial conidia* 18–36 × 3–5 μm ($\bar{x}$ = 28 × 4 μm, *n* = 20), fusiform, straight to slightly curved, sometimes with constricted septa, apical cell blunt to papillate, basal cell barely to distinctly notched, 1–3-septate, hyaline. *Sporodochia*, *Chlamydospores* and *Microconidia* not observed.

Culture characteristics: Colonies on PDA reaching 67–74 mm diameter after a week at 28 °C, white with pale yellow in the center, cottony to floccose, aerial mycelia dense, lobate margin. Reverse white cream.

Material examined: Thailand, Chiang Rai Province, Phan District, Mueang Phan Sub-district (19.48468° N, 99.719868° E), on the panicle of *Oryza sativa*, 9 November 2021, Nootjarin Jungkhun, (NS29-1 = MFLU 25-0035); (living culture MFLUCC 24-0630).

GenBank numbers: *cmdA* = PV297812, *rpb*2 = PV394828, *tef*1-α = PV394833.

Notes: *Fusarium hainanense* was first described by Wang et al. [74], obtained from a stem of *Oryza* sp., and represents a phylo-species within the FIESC. In this study, the isolate MFLUCC 24-0630 has been identified as *F. hainanense* according to both morphology and phylogeny analysis. This species has been reported in several tropical and subtropical countries [74]. However, it has not been previously recorded in Thailand. Hence, we introduce *F. hainanense* as a new geographical record for Thailand.

***Fusarium kotabaruense*** Maryani, Sand. Den., L. Lombard, Kema & Crous, Persoonia 43: 65 (2019). (Figure 12).

Index Fungorum Number: IF828964

*Saprobic* on *Oryza sativa*. *Conidiophores* borne on aerial mycelia, septate, proliferating percurrently, usually bearing terminal phialides, irregularly branched, smooth- and thin-walled, hyaline. *Conidiogenous cells* 17.5–30 × 4–7 μm ($\bar{x}$ = 22 × 4.5 μm, *n* = 20), monophialidic, subulate to subcylindrical, smooth- and thin-walled, hyaline. *Aerial conidia* 12–36 × 5.5–8 μm ($\bar{x}$ = 29 × 7 μm, *n* = 30), straight to slightly curved, slightly falcate, apical cells gently blunt, basal cells indistinct or foot-shaped, 1–5-septate, hyaline. *Sporodochia* and *Chlamydospores* not observed.

Culture characteristics: Colonies on PDA reaching 67–70 mm diameter after a week at 28 °C, dull white to rose buff, medium-dense, cottony, radiate. Reverse yellowish white with pale orange in the center.

Material examined: Thailand, Chiang Rai Province, Mueang Chiang Rai District, Huai Sak Sub-district (19.781206° N, 99.921459° E), on the panicle of *Oryza sativa*, 17 December 2021, Sahar Absalan, (HS69-1 = MFLU 25-0037); (living culture MFLUCC 24-0633).

GenBank numbers: *cmdA* = PV297813, *rpb*2 = PV394831, *tef*1-α = PV394836.

Notes: According to the phylogram (Figure 3), isolate MFLUCC 24-0633 was identified as *Fusarium kotabaruense*, positioned basally to the FIESC. Subsequent studies, however, indicated that this species is more appropriately placed within the *Fusarium camptoceras* species complex (FCSC) [75,76]. *Fusarium kotabaruense* was first reported from the infected pseudostem of *Musa* sp. var. Pisang Hawa in Indonesia [77]. This study documents the first known case of *F. kotabaruense* on rice and a new geographical record for Thailand.

***Fusarium mianyangense*** S.L. Han, M.M. Wang & L. Cai, Stud. Mycol. 104: 131 (2023). (Figure 13).

Index Fungorum Number: IF847026

*Saprobic* on *Oryza sativa*. *Conidiophores* borne on sporodochia, verticillately branched, proliferating percurrently, smooth- and thin-walled, hyaline. *Conidiogenous cells* 8–17.5 × 3–3.5 μm ($\bar{x}$ = 11 × 3.5 μm, *n* = 20), monophialidic, subulate to subcylindrical, smooth- and thin-walled, hyaline. *Sporodochial conidia* 18–46.5 × 3–5 μm ($\bar{x}$ = 26 × 4 μm, *n* = 30), straight to slightly curved, falcate, apical cell blunt and sometimes papillate, basal cell barely notched, mostly 1–4-septate, hyaline. *Sporodochia* pale orange, formed on the surface of carnation leaves. *Chlamydospores* not observed.

Culture characteristics: Colonies on PDA reaching 86–90 mm diameter after a week at 28 °C, yellowish white, velvety to floccose, slightly convex, dense, entire margin. Reverse yellowish white with pale orange in the center.

Material examined: Thailand, Chiang Rai Province, Phan District, Mueang Phan Sub-district (19.528477° N, 99.74594° E), on the seedling of *Oryza* sativa, 26 November 2021, Nootjarin Jungkhun, (NS08-2a = MFLU 25-0031); (living culture MFLUCC 24-0625).

GenBank numbers: *cmdA* = PV297814, *rpb*2 = PV394829, *tef*1-α = PV394834.

Notes: According to the results of phylogenetic analyses, the isolate MFLUCC 24-0625 is closely related to *Fusarium mianyangense*, with 90% ML and 1.00 PP support (Figure 3). Morphologically, our isolate is similar to the ex-type strain in the size and shape of conidiogenous cells and conidia, with the exception of the chlamydospores, which were not observed in our study. Consequently, based on both morphological and phylogenetic evidence, the isolate MFLUCC 24-0625 was identified as *F. mianyangense*.

***Fusarium oryzigenum*** S. Absalan, S. Lumyong & K. D. Hyde, sp. nov. (Figure 14)

Index Fungorum Number: IF903756

Etymology: Name refers to the host genus *Oryza*, from which it was isolated.

*Saprobic* on *Oryza sativa*. *Conidiophores* borne on aerial mycelia, septate, proliferating percurrently, bearing terminal or lateral phialides, irregularly and verticillately branched, smooth- and thin-walled, hyaline. *Conidiogenous cells* 15–27 × 2.5–3.5 μm ($\bar{x}$ = 20 × 3 μm, *n* = 20), monophialidic, subulate to subcylindrical, periclinal thickening inconspicuous, smooth- and thin-walled, hyaline. *Aerial conidia* 6–10 × 2.5–3.5 μm ($\bar{x}$ = 8 × 3 μm, *n* = 20), ellipsoidal to clavate, usually single on the tips of phialides, aseptate, hyaline. *Sporodochia*, *Chlamydospores* and *Macroconidia* not observed.

Culture characteristics: Colonies on PDA reaching 54–58 mm diameter after a week at 28 °C, white with pale saffron in the center, floccose, radiate, aerial mycelia dense, margin irregular or entire, filiform. Reverse pale orange.

Material examined: Thailand, Chiang Rai Province, Phan District, Mueang Phan Sub-district (19.528477° N, 99.74594° E), on the stem of *Oryza sativa*, 28 June 2022, Sahar Absalan, (PA159 = MFLU 25-0039); (ex-type, living culture MFLUCC 24-0637).

GenBank numbers: *cmdA* = PV297815, *rpb*2 = PV394844, *tef*1-α = PV394840.

Notes: Our phylogenetic analyses (Figure 1) revealed that *Fusarium oryzigenum* is located at a distinct branch, forming a well-supported lineage (100% ML and 1.00 PP) from the clade of *Fusarium andiyazi*. *Fusarium oryzigenum* is closely related to *F. andiyazi* (CBS 119857), but it differs by 15 base pairs in the three-locus combined dataset (*cmdA*+*rpb*2+*tef*1-α). Morphologically, this species can be distinguished from *F. andiyazi* by the absence of macroconidia and chlamydospores (not observed in our collection), the length of monophialides (15–27 × 2.5–3.5 μm in *Fusarium oryzigenum* vs. 14–42 × 2–3.5 μm in *F. andiyazi*), and the proliferation of microconidia (borne singly on the tips of phialides in *Fusarium oryzigenum* vs. borne in chains or false heads in *F. andiyazi*) [78]. Therefore, herein we introduce *Fusarium oryzigenum* as a new species, based on morphological examination and phylogenetic analysis.

***Fusarium sacchari*** (E.J. Butler) W. Gams, *Cephalosporium-artige Schimmelpilze* (Stuttgart): 218 (1971). (Figure 15).

Index Fungorum Number: IF314221

*Saprobic* on *Oryza sativa*. *Conidiophores* borne on aerial mycelia, septate, proliferating percurrently, bearing terminal or lateral phialides, irregularly branched, smooth- and thin-walled, hyaline. *Conidiogenous cells* 6.5–27 × 2.5–4 μm ($\bar{x}$ = 19 × 3.5 μm, *n* = 20), mono- and polyphialidic, subulate to subcylindrical, smooth- and thin-walled, hyaline. *Aerial conidia* 6–10 × 2.5–3 μm ($\bar{x}$ = 7 × 2.7 μm, *n* = 30), ovoid to ellipsoid, straight to slightly curved, hyaline. *Sporodochia* and *Macroconidia* not observed. *Chlamydospores* 5.2–7 μm diameter, abundant, subglobose to ovoid, smooth- or rough-walled, terminal or intercalary, solitary, hyaline.

Culture characteristics: Colonies on PDA reaching 80–83 mm diameter after a week at 28 °C, white with pale pink in the center, medium-dense, cottony. Reverse pink-white in the center, white at the margin.

Material examined: Thailand, Chiang Rai Province, Mueang Chiang Rai District, Tha Sut Sub-district (20.060626° N, 99.850604° E), on the panicle of *Oryza sativa*, 17 December 2021, Sahar Absalan, (TS108a = MFLU 25-0038); (living culture MFLUCC 24-0635).

GenBank numbers: *cmdA* = N/A, *rpb*2 = PV394832, *tef*1-α = PV394841.

Notes: Based on the morphological and molecular data from our study, the isolate MFLUCC 24-0635 was identified as a representative of *Fusarium sacchari*, with 96% ML and 1.00 PP support (Figure 1). According to previous studies, *F. sacchari* is one of the most prevalent pathogens affecting a variety of crops globally, including rice, in which it causes bakanae and spikelet rot diseases [79,80,81,82].

***Fusarium sulawesiense*** Maryani, Sand.-Den., L. Lombard, Kema & Crous [as “*sulawense*”], Persoonia 43: 65 (2019). (Figure 16).

Index Fungorum Number: IF314221

*Saprobic* on *Oryza sativa*. *Conidiophores* borne on aerial mycelia, septate, proliferating percurrently, bearing terminal or lateral phialides, irregularly branched, smooth- and thin-walled, hyaline. *Conidiogenous cells* 14–20 × 2.5–4 μm ($\bar{x}$ = 16 × 3 μm, *n* = 15), mono- and polyphialidic, subulate to subcylindrical, smooth- and thin-walled, hyaline. *Conidia* 10–28 × 3.5–5 μm ($\bar{x}$ = 20 × 4.5 μm, *n* = 20), sometimes obovoid to ellipsoidal, straight to slightly curved, apical cells papillate, basal cells indistinct or foot-shaped, 1–3-septate, hyaline. *Sporodochia* and *Chlamydospores* not observed.

Culture characteristics: Colonies on PDA reaching 69–72 mm diameter after a week at 28 °C, pale buff, aerial mycelia medium-dense, immersed mycelia at the margin, radiate, cottony. Reverse pale yellow.

Material examined: Thailand, Chiang Rai Province, Phan District, Mueang Phan Sub-district (19.48468° N, 99.719868° E), on the panicle of *Oryza sativa*, 9 November 2021, Nootjarin Jungkhun, (NS28-4 = MFLU 25-0034); (living culture MFLUCC 24-0629).

GenBank numbers: *cmdA* = PV297816, *rpb*2 = PV394830, *tef*1-α = PV394835.

Notes: Our isolate MFLUCC 24-0629 clustered with the type (InaCC F940) and other strains of *Fusarium sulawesiense*, with 89% ML and 0.87 PP support in the phylogenetic analysis of the FIESC (Figure 3). Infected pseudostem of *Musa acuminata* var. Pisang Cere, collected in Indonesia, was documented as the type substrate of *F. sulawesiense* [77]. The first association of this species with rice was reported in China by Wang et al. [74]

***Ochronectria thailandica*** Q.J. Shang & K.D. Hyde, Fungal Diversity 78: (84) (2016). (Figure 17).

Index Fungorum Number: IF551918

*Saprobic* on *Oryza sativa*. *Conidiophores* 36–84 × 2–3.5 μm ($\bar{x}$ = 63 × 3 μm, *n* = 15), erect, straight, unbranched, septate, slightly inflated at base, smooth-walled, hyaline. *Conidiogenous cells* monophialidic, terminally proliferating, subulate or cylindrical, producing a single conidium, smooth-walled, hyaline. *Conidia* 3.8–7.5 × 3–4 μm ($\bar{x}$ = 6 × 3.5 μm, *n* = 25), aseptate, subglobose to ellipsoidal, thin- and smooth-walled, hyaline.

Culture characteristics: Colonies on PDA reaching 40–45 mm diameter after a week at 28 °C, greyish salmon with white center, cottony to felty, aerial mycelia moderate, margin entire. Reverse pale orange with pale greyish white margin.

Material examined: Thailand, Chiang Rai Province, Mueang Chiang Rai District, Huai Sak Sub-district (19.781206° N, 99.921459° E), on the stem of *Oryza sativa*, 17 December 2021, Sahar Absalan, (HS59-1 = MFLU 25-0042); (living culture MFLUCC 24-0632).

GenBank numbers: ITS = PV243865, LSU = PV243885, *rpb*2 = PV297809, *tef*1-α = PV297807.

Notes: Based on the phylogenetic analyses of concatenated ITS, LSU, *rpb*2, and *tef*1-α sequence data, our strain (MFLUCC 24-0632) clustered with *Ochronectria thailandica*, with 95% ML and 1.00 PP support (Figure 4). It is worth noting that previously reported strains of *O. thailandica* have been identified in their sexual form [56,83]. This study, however, is the first to document its asexual morph. Additionally, it marks the first recorded association of *O. thailandica* with rice.

***Sarocladium bactrocephalum*** (W. Gams) Summerb., Stud. Mycol. 68: 158 (2011). (Figure 18).

Index Fungorum Number: IF519590

*Saprobic* on *Oryza sativa*. *Conidiophores* 23–40 × 2–3 μm ($\bar{x}$ = 36 × 2.5 μm, *n* = 20), erect, straight to flexuous, sometimes slightly bent, arising from aerial mycelium or radiating out from the mycelial coils, unbranched or basitonously branched, 1–2-septa in base, smooth-walled, hyaline. *Phialides* 12–28 × 1–2.5 μm ($\bar{x}$ = 19 × 1.5 μm, *n* = 20), lateral or terminal, subulate or cylindrical, sometimes constricted at the base, smooth-walled, hyaline. *Conidia* 4–6.5 × 1.5–2.5 μm ($\bar{x}$ = 5 × 2 μm, *n* = 40), aseptate, straight, short cylindrical, rounded at both ends, thin- and smooth-walled, 2-guttulate, hyaline. *Crystals* present. *Chlamydospores* not observed.

Culture characteristics: Colonies on PDA reaching 35–37 mm diameter after a week at 28 °C, pale salmon, slow growing, hairy, flat, aerial mycelia sparse, margin entire. Reverse white with pale orange in the center.

Material examined: Thailand, Chiang Rai Province, Mueang Chiang Rai District, Mae Yao Sub-district (19.968012° N, 99.772583° E), on the panicle of *Oryza sativa*, 17 December 2021, Sahar Absalan, (MS89 = MFLU 25-0041); (living culture MFLUCC 24-0634).

GenBank numbers: ITS = PV243886, LSU = PV243904.

Notes: Based on the results of phylogenetic analyses, our isolate was grouped together with the type strain of *Sarocladium bactrocephalum* (CBS 749.69), with 99% ML and 1.00 PP support (Figure 5). This species was formerly known as *Acremonium bactrocephalum* and was later transferred to *Sarocladium* by Summerbell et al. [40]. According to Gams [84], *S. bactrocephalum* shares a close relationship with *S. strictum* but can be differentiated by its distinctively long and narrow conidia. Additionally, it is molecularly distinguishable through LSU sequence analysis. In this study, we provided the first association of this species with rice as a new host record.

***Sarocladium oryzae*** (Sawada) W. Gams & D. Hawksw., Kavaka 3: 58 (1976) [1975]. (Figure 19).

Index Fungorum Number: IF323106

*Saprobic* on *Oryza sativa*. *Conidiophores* 30–80 × 1.5–3 μm ($\bar{x}$ = 43 × 2 μm, *n* = 25), erect, straight or irregularly curved and flexuose, arising from aerial mycelium, unbranched or sometimes basitonously or verticillately branched, occasionally warted near the base, 1–2-septa in base or middle, smooth-walled, hyaline. *Phialides* 20–45 × 1–2 μm ($\bar{x}$ = 43 × 1.5 μm, *n* = 30), lateral or terminal, subulate or cylindrical, thick- and smooth-walled, hyaline. *Conidia* 4.5–7 × 1.5–2.5 μm ($\bar{x}$ = 5.5 × 2 μm, *n* = 40), aseptate, straight, cylindrical, with rounded ends, thin- and smooth-walled, arranged in slimy heads, hyaline. *Chlamydospores* not observed.

Culture characteristics: Colonies on PDA reaching 90 mm diameter after a week at 28 °C, dull white to pale orange, cottony to fluffy, arised, aerial mycelia dense, margin entire. Reverse pale orange with apricot orange in the center.

Material examined: Thailand, Chiang Rai Province, Phan District, Mueang Phan Sub-district (19.528477° N, 99.74594° E), on the panicle of *Oryza sativa*, 26 November 2021, Nootjarin Jungkhun, (NS02-1 = MFLU 25-0040); (living culture MFLUCC 24-0627).

GenBank numbers: ITS = PV243887, LSU = PV243905.

Notes: *Sarocladium oryzae*, initially identified as *Acrocylindrium oryzae* [85], is the type species of *Sarocladium*. In this study, the strain MFLUCC 24-0627 was identified as *Sarocladium oryzae*, based on the similarity in the morphological characteristics and phylogenetic analysis (Figure 5).

***Trichothecium roseum*** (Pers.) Link, Mag. Gesell. naturf. Freunde, Berlin 3(1–2): 18 (1809). (Figure 20).

Index Fungorum Number: IF152448

*Saprobic* on *Oryza sativa*. *Conidiophores* 67–117 × 4–5.5 μm ($\bar{x}$ = 85 × 5 μm, *n* = 15), erect, straight, occasionally branched, septate, rough-walled, hyaline. *Conidiogenous cells* monoblastic, terminally proliferating, hyaline. *Conidia* 10–23.5 × 7–13.5 μm ($\bar{x}$ = 16.5 × 11 μm, *n* = 30), initially aseptate, two-celled in mature conidia, sometimes with constricted septa, ellipsoid to pyriform, apical cell rounded, basal cell blunt, smooth-walled, hyaline.

Culture characteristics: Colonies on PDA reaching 64–66 mm diameter after a week at 28 °C, initially white, becoming rose buff, with concentric rings, powdery or granular, flat, margin entire. Reverse pale buff.

Material examined: Thailand, Chiang Rai Province, Mueang Chiang Rai District, Tha Sut Sub-district (20.060626° N, 99.850604° E), on the panicle of *Oryza sativa*, 17 December 2021, Sahar Absalan, (TS101 = MFLU 25-0043); (living culture MFLUCC 24-0636).

GenBank numbers: ITS = PV243909, *tef*1-α = PV297805.

Notes: Morphology and molecular analyses confirmed that strain MFLUCC 24-0636 belongs to *Trichothecium roseum*, with 100% ML and 1.00 PP support (Figure 6). *Trichothecium roseum* is a significant postharvest pathogen with a global presence, responsible for pink rot and white stain diseases in various fruit crops. This fungus commonly inhabits soil and can become pathogenic under favorable conditions [86,87,88,89,90,91,92,93]. In rice, *T. roseum* has been reported during storage, and previous studies in Thailand have documented its occurrence in both rice and sorghum [94,95].

***Waltergamsia fusidioides*** (Nicot) L.W. Hou, L. Cai & Crous, in Hou, Giraldo, Groenewald, Rämä, Summerbell, Huang, Cai & Crous, Stud. Mycol. 105: 141 (2023) (Figure 21).

Index Fungorum Number: IF845886

*Saprobic* on *Oryza sativa*. *Conidiophores* 22–47 × 1.4–2 μm ($\bar{x}$ = 31 × 1.5 μm, *n* = 10), solitary, erect, straight or curved, aseptate, usually reduced to single phialides, smooth-walled, hyaline. *Phialides* lateral or terminal, subulate, smooth-walled, hyaline. *Conidia* 4.2–6.7 × 2.5–3.5 μm ($\bar{x}$ = 5.7 × 3 μm, *n* = 30), aseptate, thin- and smooth-walled, ovoid or fusoid, sometimes arranged in long chains, hyaline.

Culture characteristics: Colonies on PDA reaching 30–33 mm diameter after a week at 28 °C, white, felty, slightly raised, radially folded, margin crenate. Reverse greyish white.

Material examined: Thailand, Chiang Rai Province, Mueang Chiang Rai District, Tha Sut Sub-district (20.060626° N, 99.850604° E), on the panicle of *Oryza sativa*, 17 December 2021, Sahar Absalan, (M2S224 = MFLU 25-0044); (living culture MFLUCC 24-0638).

GenBank numbers: ITS = PV243954, LSU = PV243955, *rpb*2 = PV297808, *tef*1-α = PV297806.

Notes: Based on the molecular data, the strain MFLUCC 24-0638 was identified as *Waltergamsia fusidioides*, with 100% ML and 1.00 PP support (Figure 7). *Waltergamsia fusidioides* was recently reclassified by Hou et al. [96] as a new combination for the ex-type culture of *Paecilomyces fusidioides*, now considered its basionym. Morphological examination revealed that this species typically produces two types of conidia: globose and fusoid [84]. In our study, however, only one type of conidia was observed, which may be attributed to differences in substrate. This study reports the first documentation of *W. fusidioides* associated with rice and represents its new geographical record for Thailand.

## 4. Discussion

This study reports on nine *Fusarium* species along with five species in four other genera from rice. Nine *Fusarium* species were collected in total, belonging to three species complexes: the *Fusarium fujikoroi* species complex (FFSC), the *Fusarium incarnatum*-*equiseti* species complex (FIESC), and the *Fusarium nisikadoi* species complex (FNSC). Within the FFSC, three species of *Fusarium* were identified, including two newly described species (*F. chiangraiense* and *F. oryzigenum*) and one known species (*F. sacchari*). *Fusarium sacchari* has previously been reported from various grains, including rice (*Oryza sativa*) and wild rice (*Oryza australiensis*), in Australia, India, Indonesia, and Malaysia [80,97,98,99,100,101]. Although this species has been isolated from sorghum (*Sorghum bicolor*) and orchid (*Rhynchostylis gigantea*) in Thailand, there has been no previous report of the species associated with rice [102,103].

Within the FIESC, five species were identified, including new host and geographical records. *Fusarium guilinense* was originally isolated from the leaf of *Musa nana* in China and was later reported from rice glumes as well [74,104]. Multilocus phylogenetic analysis by Han et al. [104] determined *F. bubalinum* to be synonymous with *F. guilinense,* which was first reported by Xia et al. [75] from an unknown host. Although Han et al. [104] did not record *F. guilinense* as a pathogen on rice, Pramunadipta et al. [101] reported that *F. guilinense* (synonym: *F. bubalinum*) causes sheath rot in rice. In the present study, we isolated this fungus as a saprobe from a dead rice panicle, representing a new geographical record for Thailand. *Fusarium hainanense* also represents a new geographical record for Thailand. This species has previously been found on various hosts, including *Acacia* sp., *Musa acuminata*, *Musa nana*, *Oryza australiensis*, *Oryza sativa*, and *Zea mays* [74,75,101,104]. *Fusarium hainanense* is known to cause two diseases in rice: sheath rot and spikelet rot [101,104]. Similarly, *F. mianyangense*, originally isolated from rice in China, has been documented to cause rice spikelet rot [104]. *Fusarium sulawesiense* is a fungus with a wide range of hosts and substrates, including plants and humans [74,75,77,104,105,106,107]. Its association with rice has been recorded in China, India, Indonesia, and Brazil [74,101,108], and it is confirmed as a causal agent of rice sheath rot disease [101]. In a recent study, both *F. mianyangense* and *F. sulawesiense* were isolated in Thailand as causal agents of fruit rot in muskmelon (*Cucumis melo*) [109]; thus, the current study is the second report of these species in Thailand. Another species of *Fusarium* within the FIESC identified in this study is *Fusarium kotabaruense*, previously reported from a single host, *Musa* sp. [77]. This study marks the second global record of the species and its first association with rice.

*Fusarium commune* was initially isolated from soil and peas in Denmark, as well as from a variety of other substrates, such as white pine, Douglas fir, carnation, corn, carrot, and barley, in multiple countries across the northern hemisphere [110]. Morphologically, *F. commune* closely resembles *F. oxysporum*, leading to its misidentification in earlier studies. It was later classified as a new species within the FNSC [110,111]. *Fusarium commune* is known as a pathogen that causes fusarium wilt and root rot in various plants, such as soybean, sugarcane, horseradish, and tomato [112,113,114,115]. This species has also been reported from rice in Bangladesh and Korea, where it was associated with diseases such as wilt, root rot, and bakanae [116,117]. However, in this study, it was isolated as a saprobic fungus.

Many *Sarocladium* species are commonly associated with poaceous hosts, particularly rice [42,118,119]. Among them, *S. oryzae*, the type species of the genus, has been widely reported as the causal agent of sheath rot in rice in various countries [44,119,120,121,122]. In Thailand, several studies have documented the association between *S. oryzae* and rice in different regions [54,123,124]. This study further contributes to the documentation of *S. oryzae* occurrence in rice in the northern part of Thailand. *Sarocladium bactrocephalum* has previously been isolated from *Ustilago* sp., *Brachiaria ruziziensis*, *Panicum maximum*, and the human eye [118,125]. In this study, we report it as a new host record from rice and a new geographical record for Thailand.

*Ochronectria* is a small genus within the Hypocreales, comprising only three recognized species, all previously known solely from their sexual morphs [26,126]. These fungi typically exhibit a Nectria-like sexual morph with perithecial ascomata and an Acremonium-like asexual morph [96]. In this study, we report the asexual morph of *Ochronectria thailandica* for the first time. Until now, this species had only been observed in its sexual form on unidentified submerged wood and coconut substrates [56,83]. Our discovery provides new insight into the life cycle and morphology of this rare genus.

Another new host record documented here is *Waltergamsia fusidioides*, which was isolated from the dung of antelope, decaying leaf of *Canna indica*, and an unknown substrate in the Central African Republic, Italy, and France, respectively [96].

This study expands the database of fungi associated with rice, providing valuable insights for future research on plant health, disease management, and agricultural biosecurity. The identification of *Fusarium* spp. and other species as saprobes in dead rice tissues raises important questions about their ecological roles and potential risks to rice cultivation in Thailand. Given that many *Fusarium* species are well-known pathogens, their presence as saprobes suggests that, under favorable conditions, they could transition to a pathogenic lifestyle, posing a threat to rice production [127,128]. Future research should focus on monitoring these fungal species across different growth stages of rice to determine their potential pathogenicity. Genome-based studies could provide insights into their virulence factors, while controlled pathogenicity tests would help assess their ability to infect living plants [129]. Additionally, understanding environmental triggers that influence the shift from saprobic to pathogenic phases will be crucial in developing early detection strategies and preventive measures [130,131].

These findings highlight the importance of quarantine regulations and disease management strategies [132,133]. By enhancing our knowledge of fungal diversity in rice ecosystems, this research contributes to the development of protective measures against emerging pathogens, ensuring the sustainability of rice cultivation in Thailand.

## Figures and Tables

**Figure 1 jof-11-00321-f001:**
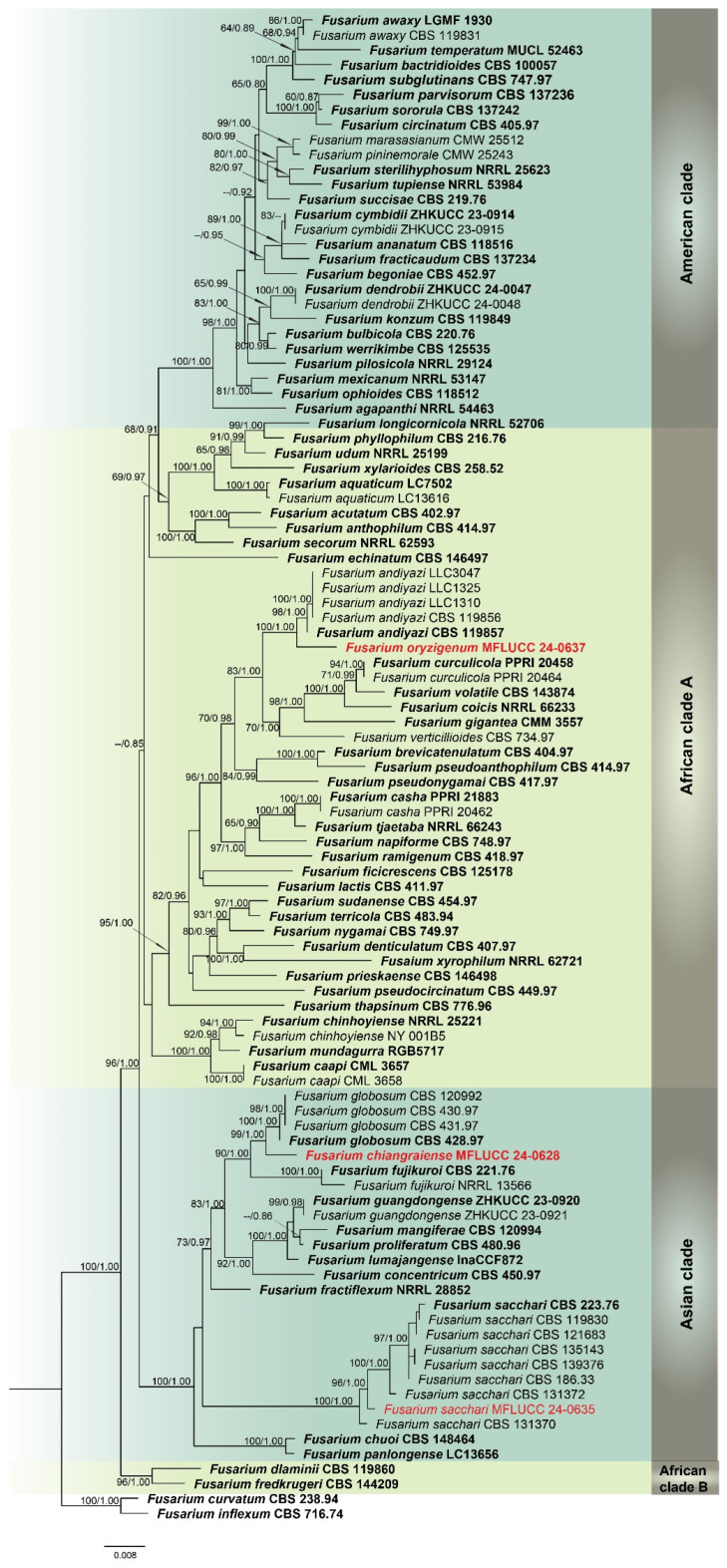
Combined phylogeny of *cmdA*, *rpb*2, and *tef*1-α gene regions of the *Fusarium fujikuroi* species complex (FFSC). The tree is rooted to *Fusarium curvatum* (CBS 238.94) and *F. inflexum* (CBS 716.74). Maximum likelihood bootstrap support values (ML) equal to or greater than 60%, and Bayesian posterior probabilities (PP) equal to or greater than 0.80, are given at the nodes (ML/PP). The isolate from the current study is highlighted in red, and type strains are indicated in bold black.

**Figure 2 jof-11-00321-f002:**
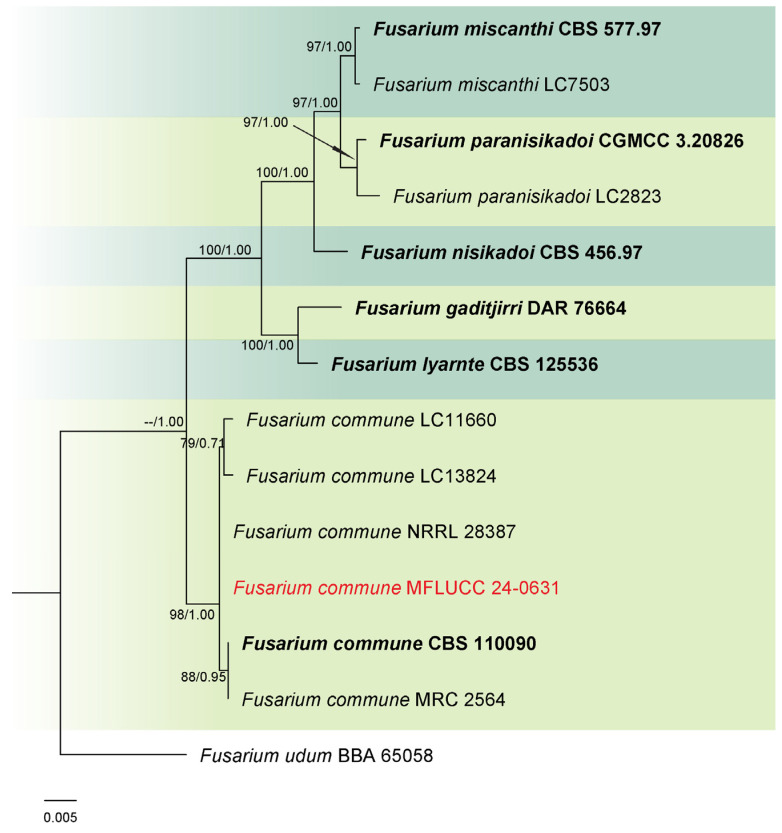
Combined phylogeny of *rpb*2 and *tef*1-α gene regions of the *Fusarium nisikadoi* species complex (FNSC). The tree is rooted to *Fusarium udum* (BBA 65058). Maximum likelihood bootstrap support values (ML) equal to or greater than 60%, and Bayesian posterior probabilities (PP) equal to or greater than 0.80, are given at the nodes (ML/PP). The isolate from the current study is highlighted in red, and type strains are indicated in bold black.

**Figure 3 jof-11-00321-f003:**
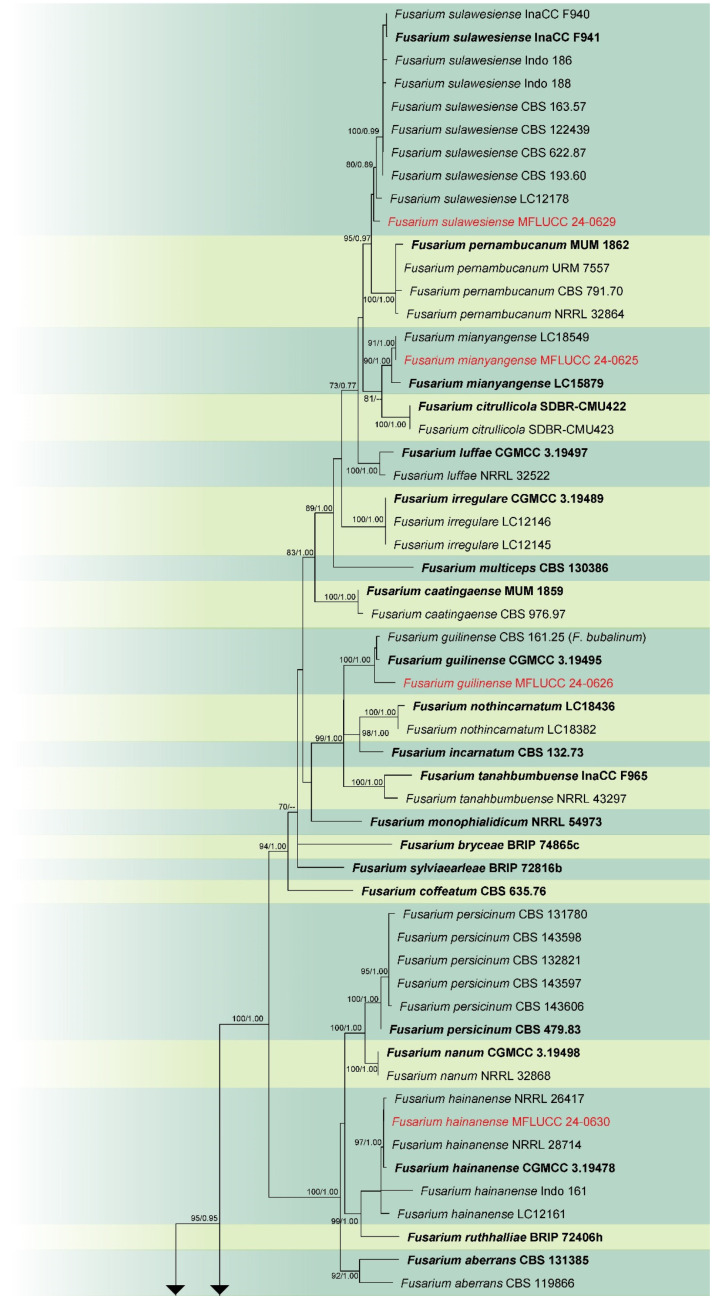
Combined phylogeny of *cmdA*, *rpb*2, and *tef*1 gene regions of the *Fusarium incarnatum*-*equiseti* species complex (FIESC). The tree is rooted to *Fusarium concolor* (CBS 961.87). Maximum likelihood bootstrap support values (ML) equal to or greater than 60%, and Bayesian posterior probabilities (PP) equal to or greater than 0.80, are given at the nodes (ML/PP). The isolate from the current study is highlighted in red, and type strains are indicated in bold black.

**Figure 4 jof-11-00321-f004:**
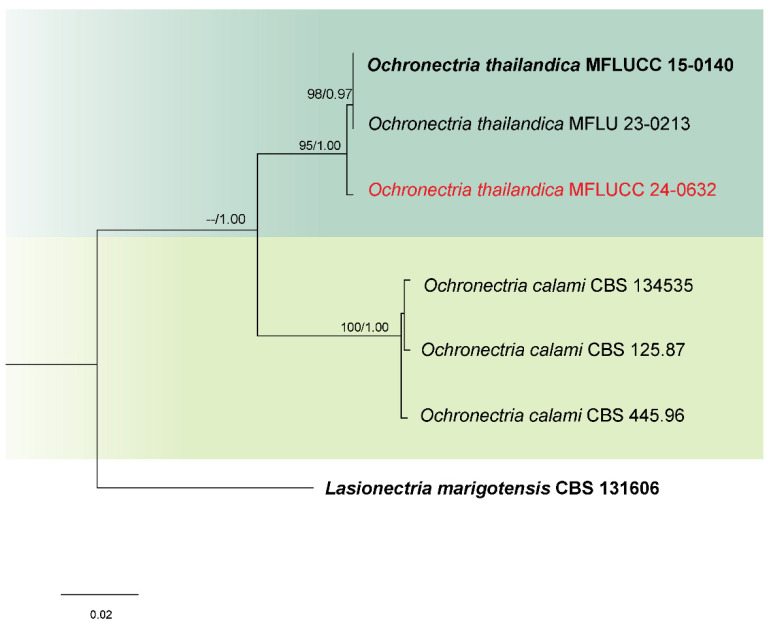
Combined phylogeny of ITS, LSU, *rpb*2, and *tef*1-α gene regions of the *Ochronectria* species. The tree is rooted to *Lasionectria marigotensis* (CBS 131606). Maximum likelihood bootstrap support values (ML) equal to or greater than 60%, and Bayesian posterior probabilities (PP) equal to or greater than 0.80, are given at the nodes (ML/PP). The isolate from the current study is highlighted in red, and type strains are indicated in bold black.

**Figure 5 jof-11-00321-f005:**
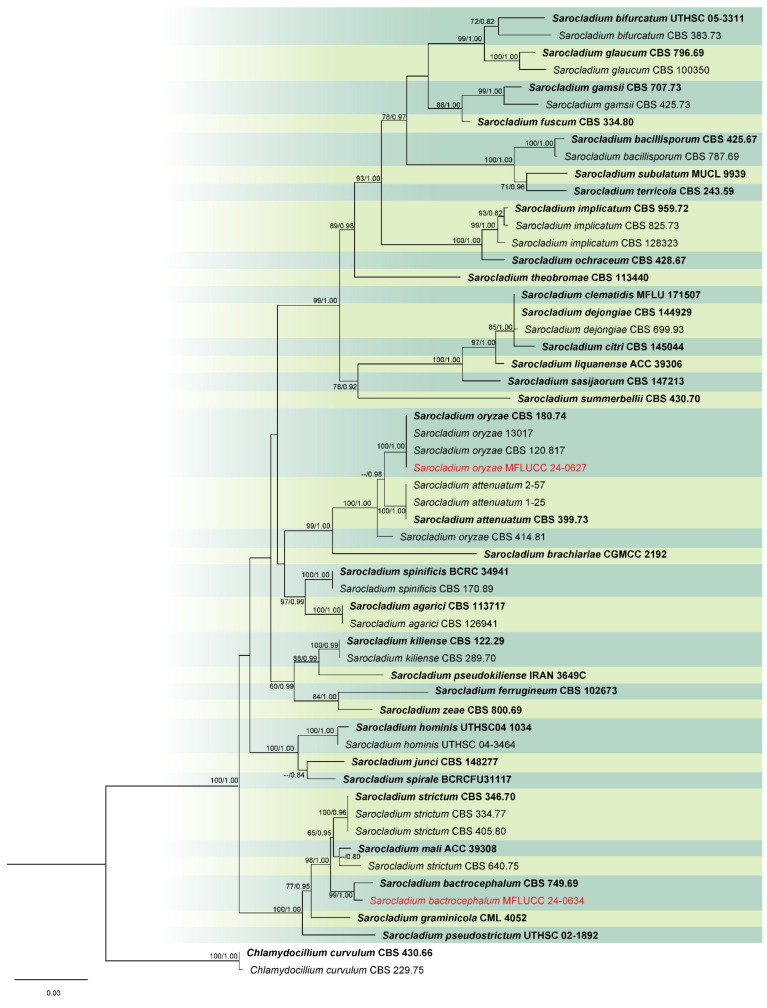
Combined phylogeny of ITS, LSU, and *act* gene regions of the *Sarocladium* species. The tree is rooted to *Chlamydocillium curvulum* (CBS 430.66) and *C. curvulum* (CBS 229.75). Maximum likelihood bootstrap support values (ML) equal to or greater than 60%, and Bayesian posterior probabilities (PP) equal to or greater than 0.80, are given at the nodes (ML/PP). The isolate from the current study is highlighted in red, and type strains are indicated in bold black.

**Figure 6 jof-11-00321-f006:**
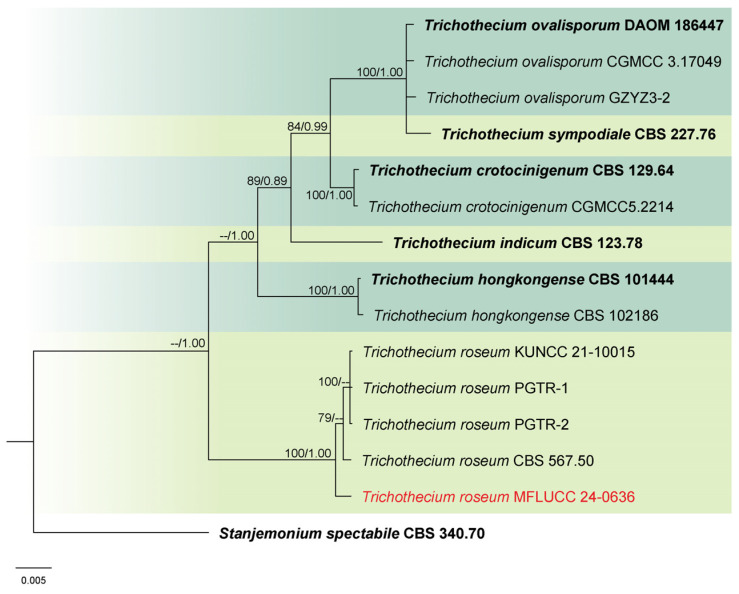
Combined phylogeny of ITS, LSU, and *tef*1-α gene regions of the *Trichothecium* species. The tree is rooted to *Stanjemonium spectabile* (CBS 340.70). Maximum likelihood bootstrap support values (ML) equal to or greater than 60%, and Bayesian posterior probabilities (PP) equal to or greater than 0.80, are given at the nodes (ML/PP). The isolate from the current study is highlighted in red, and type strains are indicated in bold black.

**Figure 7 jof-11-00321-f007:**
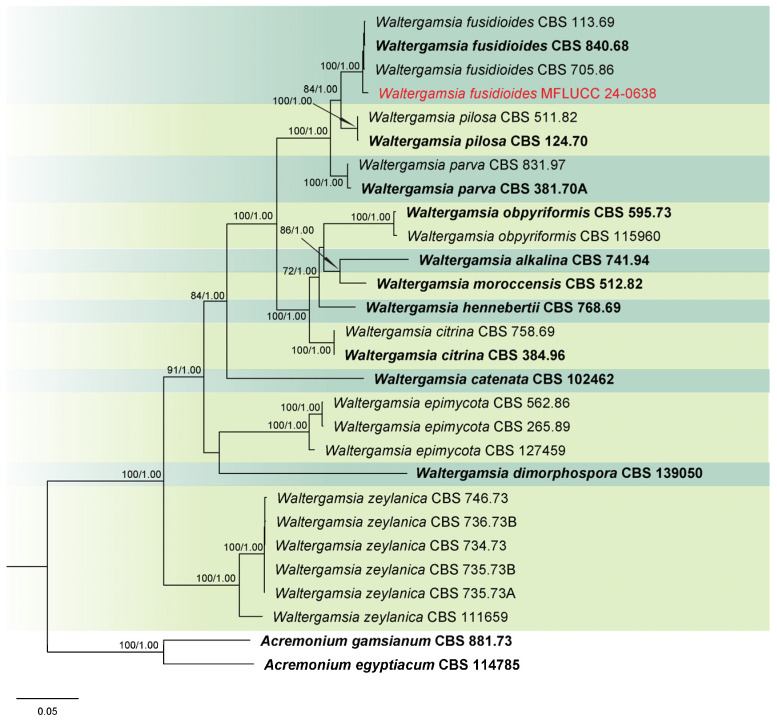
Combined phylogeny of ITS, LSU, *rpb*2, and *tef*1-α gene regions of the *Waltergamsia* species. The tree is rooted to *Acremonium egyptiacum* (CBS 114785) and *A. gamsianum* (CBS 881.73). Maximum likelihood bootstrap support values (ML) equal to or greater than 60%, and Bayesian posterior probabilities (PP) equal to or greater than 0.80, are given at the nodes (ML/PP). The isolate from the current study is highlighted in red, and type strains are indicated in bold black.

**Figure 8 jof-11-00321-f008:**
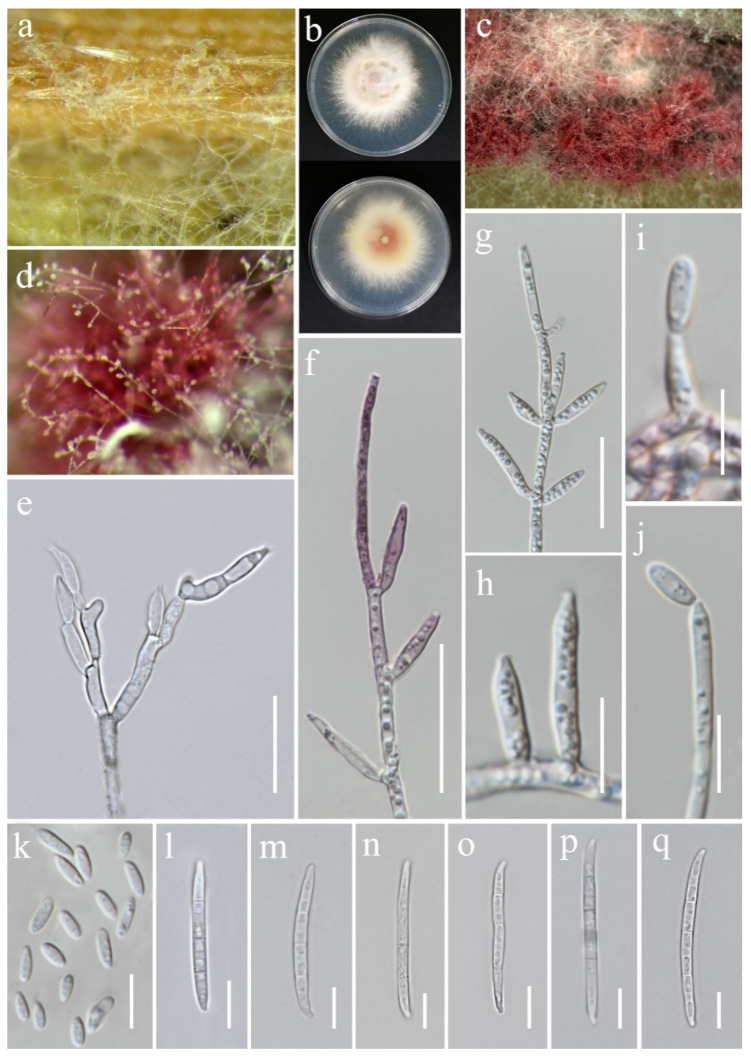
*Fusarium chiangraiense* (MFLUCC 24-0628, ex-type). (**a**) Appearance of mycelia on the panicle of *Oryza sativa*; (**b**) front and reverse of colony on PDA; (**c**,**d**) conidiophores and conidia formed on the carnation leaf; (**e**–**j**) mono- and polyphialides on aerial mycelium; (**k**) aerial microconidia; (**l**–**q**) aerial macroconidia. Scale bars: (**e**,**f**) = 20 μm; (**g**–**q**) = 10 μm.

**Figure 9 jof-11-00321-f009:**
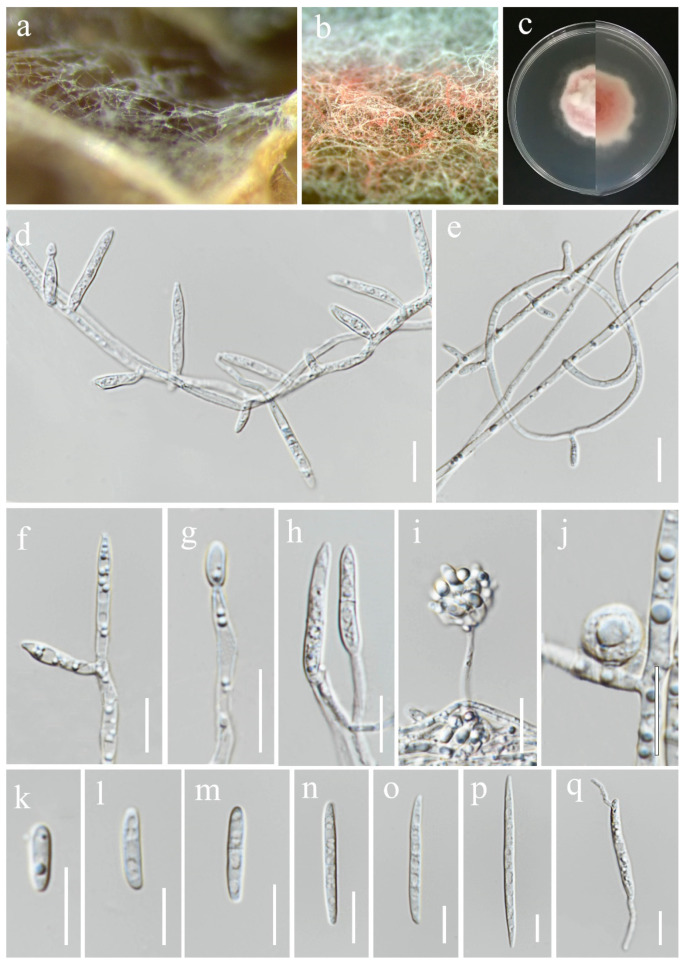
*Fusarium commune* (MFLUCC 24-0631). (**a**,**b**) Appearance of mycelia on the flag leaf of *Oryza sativa* and carnation leaves, respectively; (**c**) front and reverse of colony on PDA; (**d**–**h**) monophialides and conidia on aerial mycelium; (**i**) microconidia on false head; (**j**) chlamydospores; (**k**–**p**) aerial conidia; (**q**) microcyclic conidiation. Scale bars: (**d**–**q**) = 10 μm.

**Figure 10 jof-11-00321-f010:**
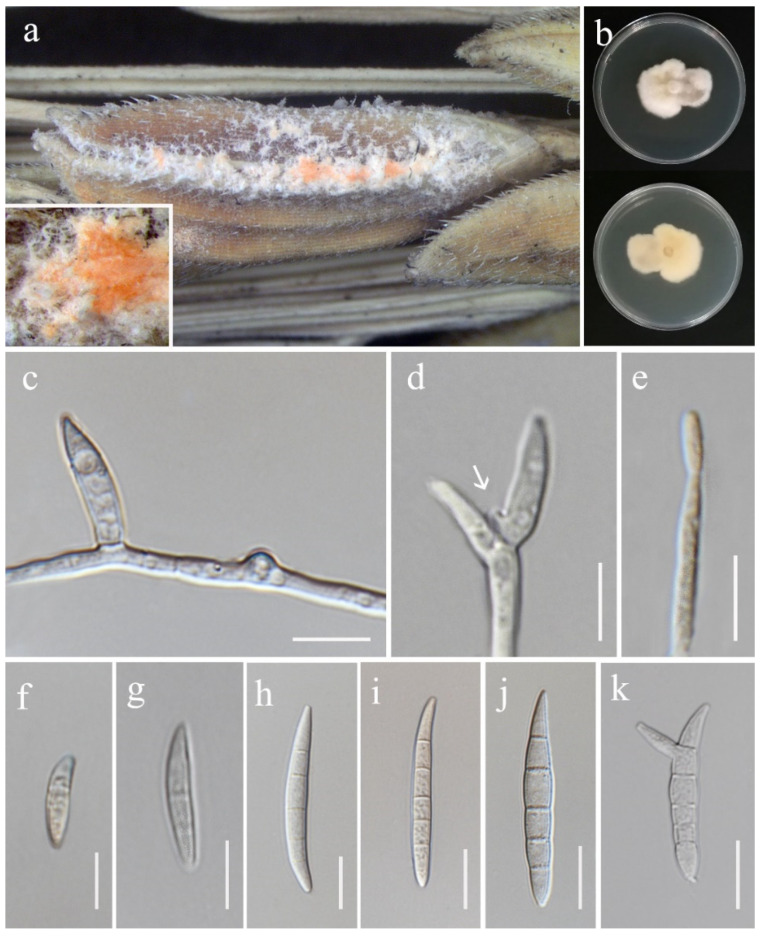
*Fusarium guilinense* (MFLU 25-0032, new geographical record). (**a**) Appearance of mycelia on the panicle of *Oryza sativa*; (**b**) front and reverse of colony on PDA; (**c**) lateral monophialide on aerial mycelium; (**d**,**e**) mono- and polyphialides on aerial mycelium (arrow pointing at phialid); (**f**–**j**) aerial conidia; (**k**) microcyclic conidiation. Scale bars: (**c**–**k**) = 10 μm.

**Figure 11 jof-11-00321-f011:**
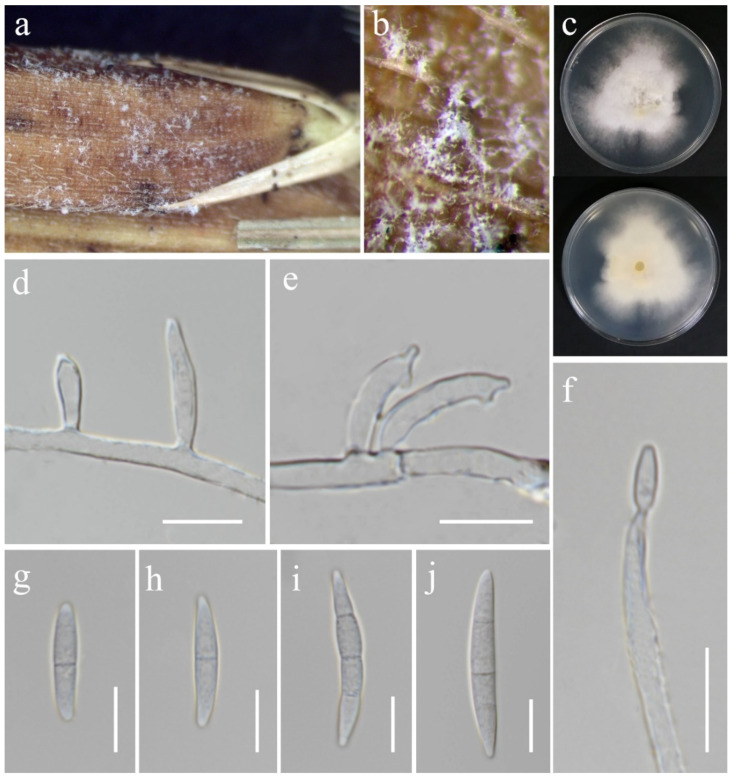
*Fusarium hainanense* (MFLU 25-0035). (**a**,**b**) Appearance of mycelia on the panicle of *Oryza sativa*; (**c**) front and reverse of colony on PDA; (**d**) monophialides on aerial mycelium; (**e**) polyphialides on aerial mycelium; (**f**) conidium produced on monophialide; (**g**–**j**) aerial conidia. Scale bars: (**d**–**j**) = 10 μm.

**Figure 12 jof-11-00321-f012:**
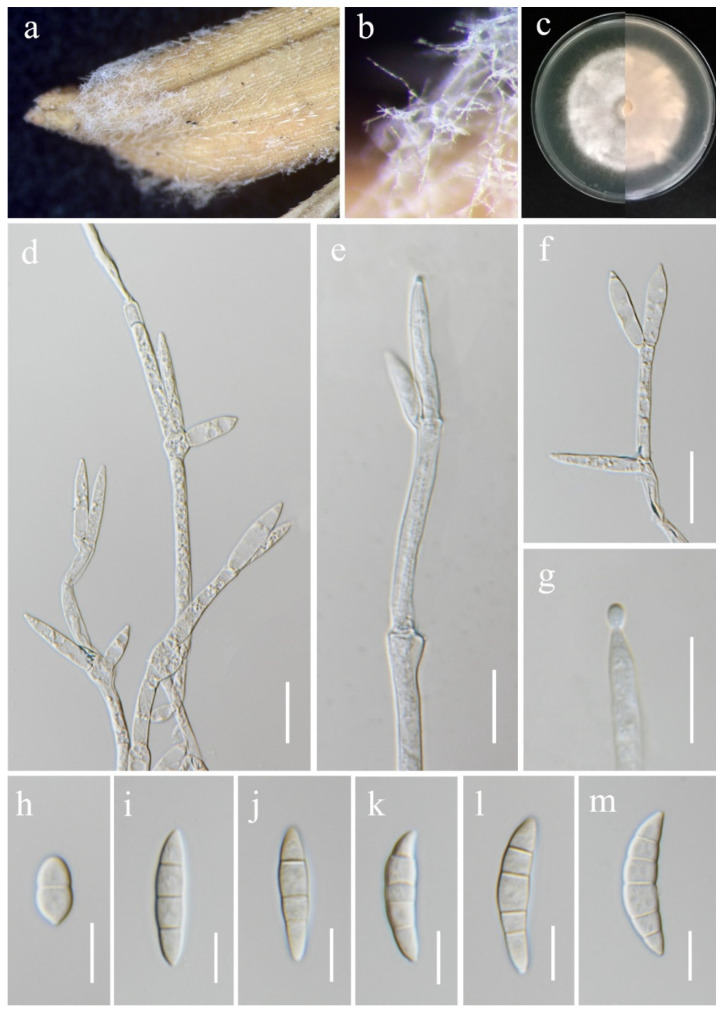
*Fusarium kotabaruense* (MFLU 25-0037, new host and geographical record). (**a**,**b**) Appearance of mycelia on the panicle of *Oryza sativa*; (**c**) front and reverse of colony on PDA; (**d**–**f**) conidiophores and conidiogenous cells; (**g**) aerial conidium produced on a monophialide; (**h**–**m**) aerial conidia. Scale bars: (**d**–**g**) = 20 μm; (**h**–**m**) = 10 μm.

**Figure 13 jof-11-00321-f013:**
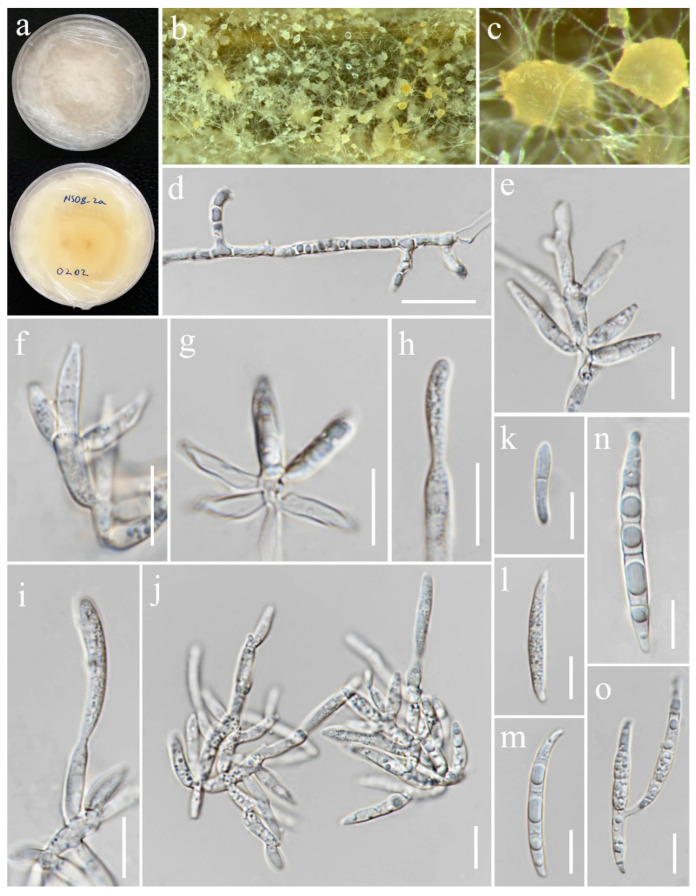
*Fusarium mianyangense* (MFLUCC 24-0625). (**a**) Front and reverse of colony on PDA; (**b**,**c**) sporodochia formed on the carnation leaf; (**d**) lateral monophialides on aerial mycelium; (**e**–**j**) monophialides and conidia formed on sporodochia; (**k**–**n**) sporodochial conidia; (**o**) microcyclic conidiation. Scale bars: (**d**) = 20 μm; (**e**–**o**) = 10 μm.

**Figure 14 jof-11-00321-f014:**
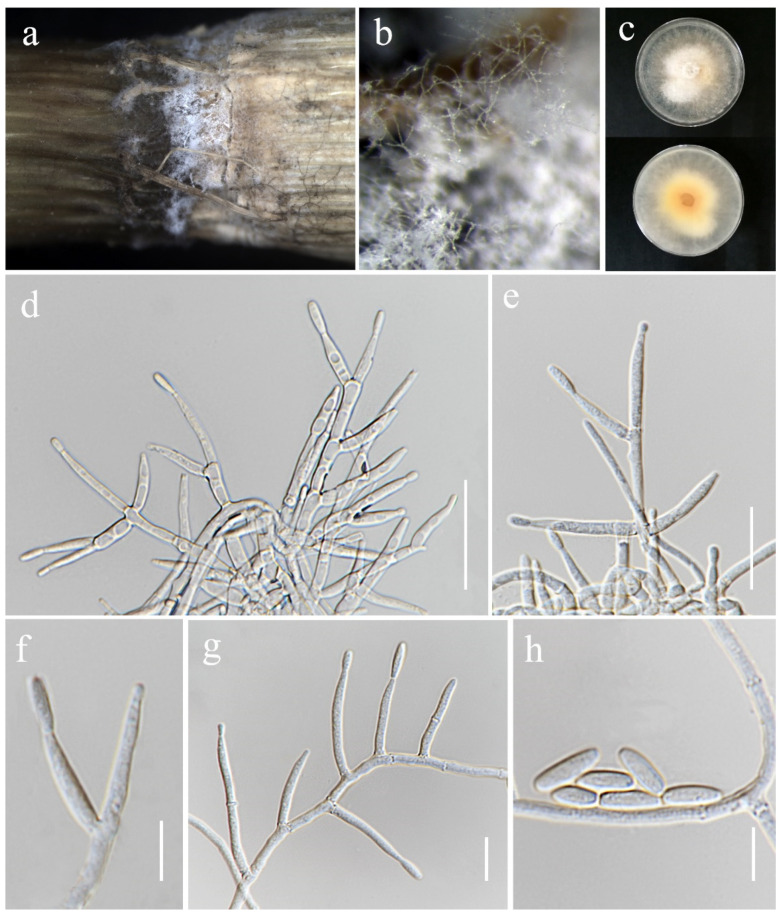
*Fusarium oryzigenum* (MFLU 25-0039, holotype). (**a**,**b**) Appearance of mycelia on the stem of *Oryza sativa*; (**c**) front and reverse of colony on PDA; (**d**–**g**) conidiophores and conidiogenous cells; (**h**) aerial conidia. Scale bars: (**d**–**h**) = 10 μm.

**Figure 15 jof-11-00321-f015:**
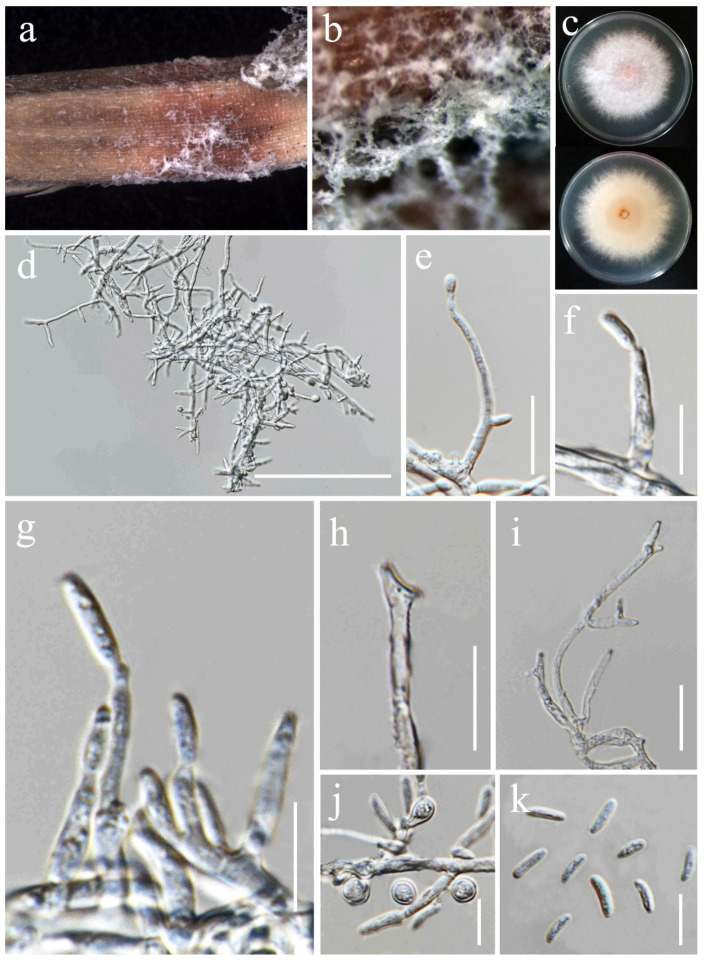
*Fusarium sacchari* (MFLU 25-0038). (**a**,**b**) Appearance of mycelia on the panicle of *Oryza sativa*; (**c**) front and reverse of colony on PDA; (**d**) conidiophores and conidiogenous cells; (**e**–**g**) aerial conidia produced on monophialides; (**h**,**i**) polyphialieds; (**j**) chlamydospores; (**k**) aerial conidia. Scale bars: (**d**) = 100 μm; (**e**,**i**) = 20 μm; (**f**–**h**,**j**–**k**) = 10 μm.

**Figure 16 jof-11-00321-f016:**
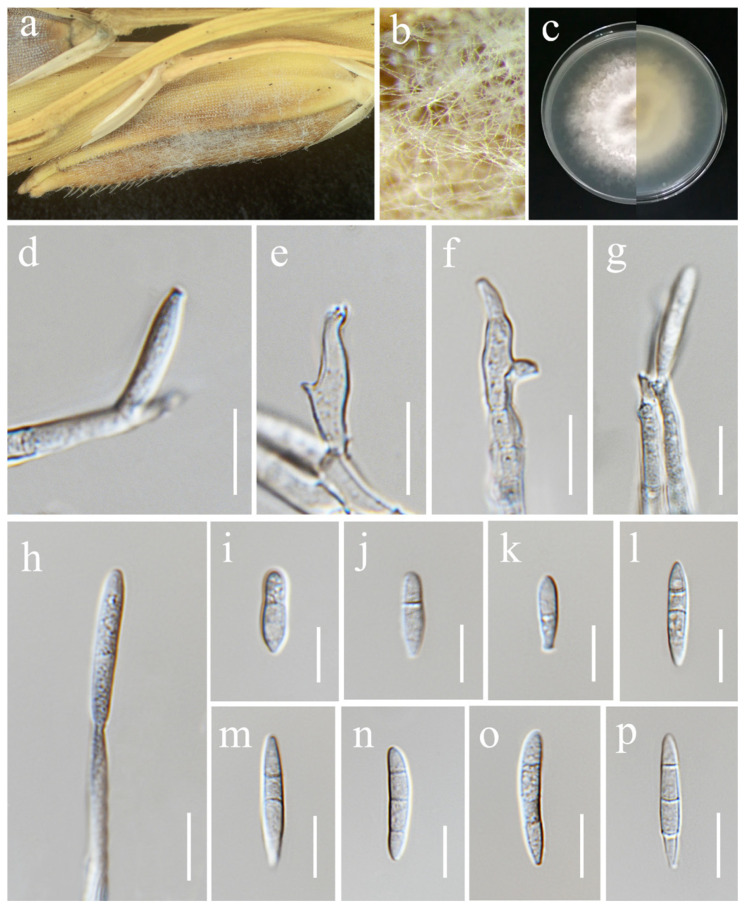
*Fusarium sulawesiense* (MFLU 25-0034). (**a**,**b**) Appearance of mycelia on the panicle of *Oryza sativa*; (**c**) front and reverse of colony on PDA; (**d**,**f**) monophialides; (**e**) polyphialides; (**g**,**h**) aerial conidia produced on mono- and polyphialides; (**i**–**p**) aerial conidia. Scale bars: (**d**–**p**) = 10 μm.

**Figure 17 jof-11-00321-f017:**
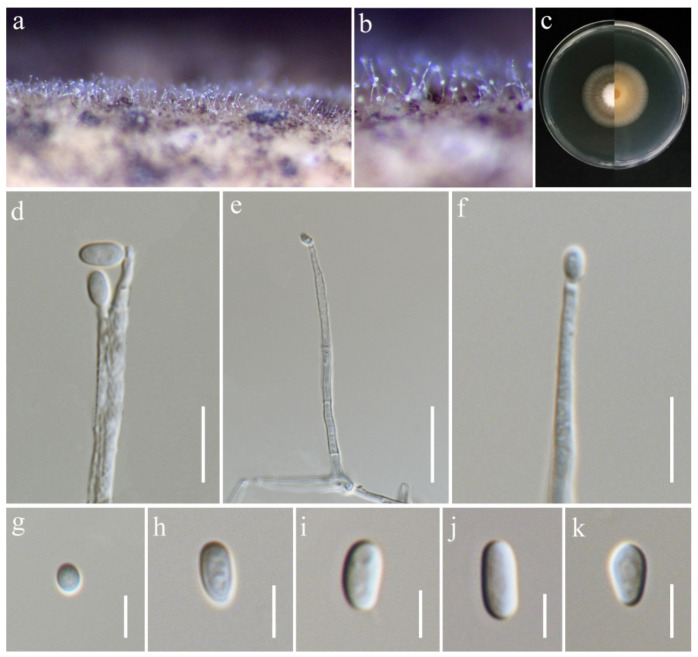
*Ochronectria thailandica* (MFLU 25-0042, new host record). (**a**,**b**) Appearance of conidiophores on the stem of *Oryza sativa*; (**c**) front and reverse of colony on PDA; (**d**–**f**) conidiophores; (**g**–**k**) conidia. Scale bars: (**d**,**e**) = 20 μm; (**f**) = 10 μm; (**g**–**k**) = 5 μm.

**Figure 18 jof-11-00321-f018:**
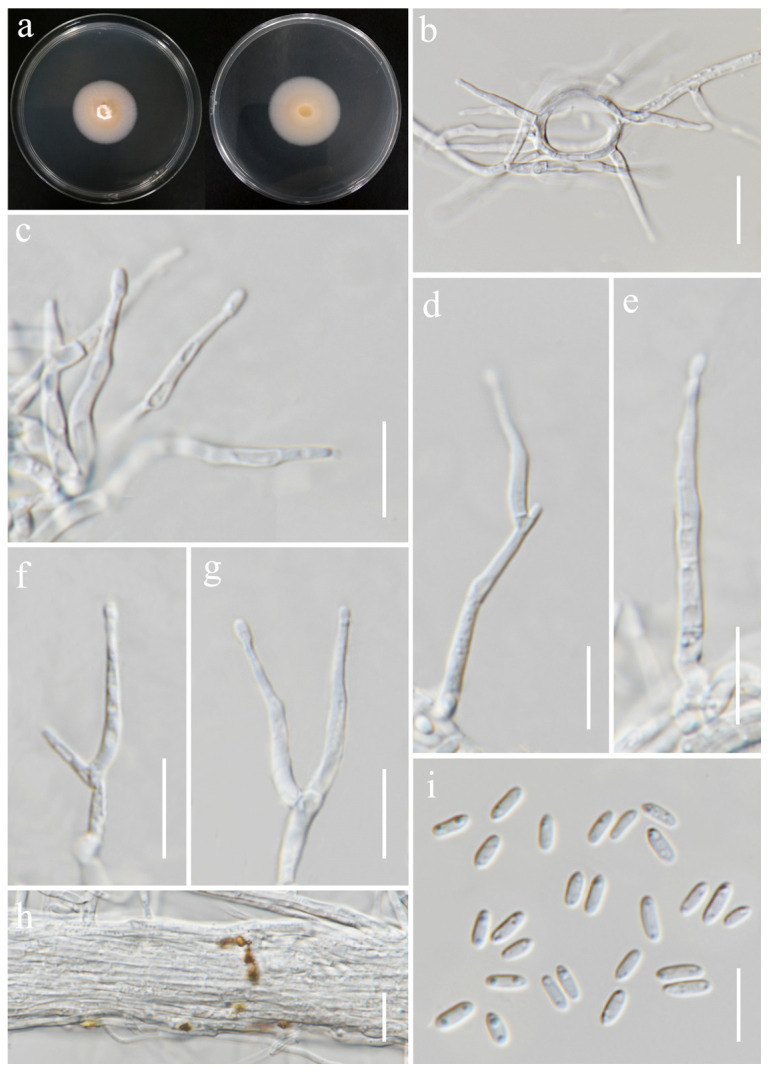
*Sarocladium bactrocephalum* (MFLUCC 24-0634, new host record). (**a**) Front and reverse of colony on PDA; (**b**) conidiophores radiating out from coils; (**c**–**g**) conidiophores; (**h**) rope of hyphae-containing crystals; (**i**) conidia. Scale bars: (**b**–**i**) = 20 μm.

**Figure 19 jof-11-00321-f019:**
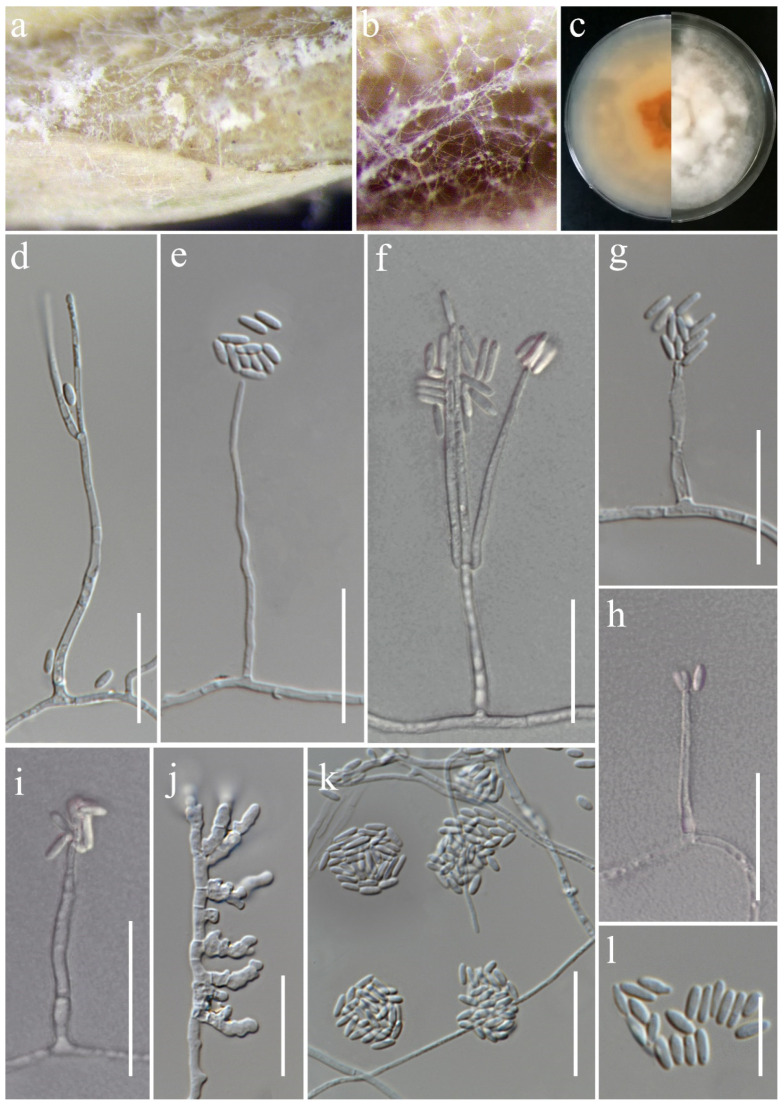
*Sarocladium oryzae* (MFLUCC 24-0627). (**a**,**b**) Appearance of mycelia on the panicle of *Oryza sativa*; (**c**) front and reverse of colony on PDA; (**d**–**i**) conidiophores; (**j**) inflated hyphae; (**k**) conidia in slimy heads; (**l**) conidia. Scale bars: (**d**–**l**) = 20 μm.

**Figure 20 jof-11-00321-f020:**
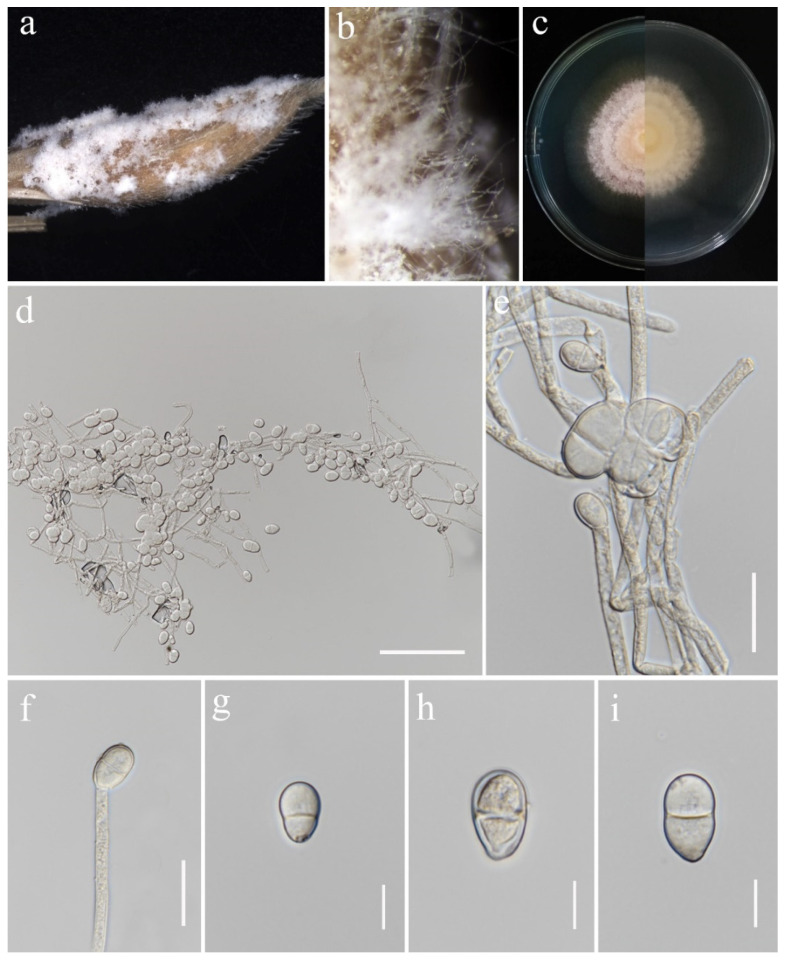
*Trichothecium roseum* (MFLU 25-0043). (**a**,**b**) Appearance of mycelia on the panicle of *Oryza sativa*; (**c**) front and reverse of colony on PDA; (**d**–**f**) conidiophores; (**g**–**i**) conidia. Scale bars: (**d**) = 100 μm; (**e**,**f**) = 20 μm; (**g**–**i**) = 10 μm.

**Figure 21 jof-11-00321-f021:**
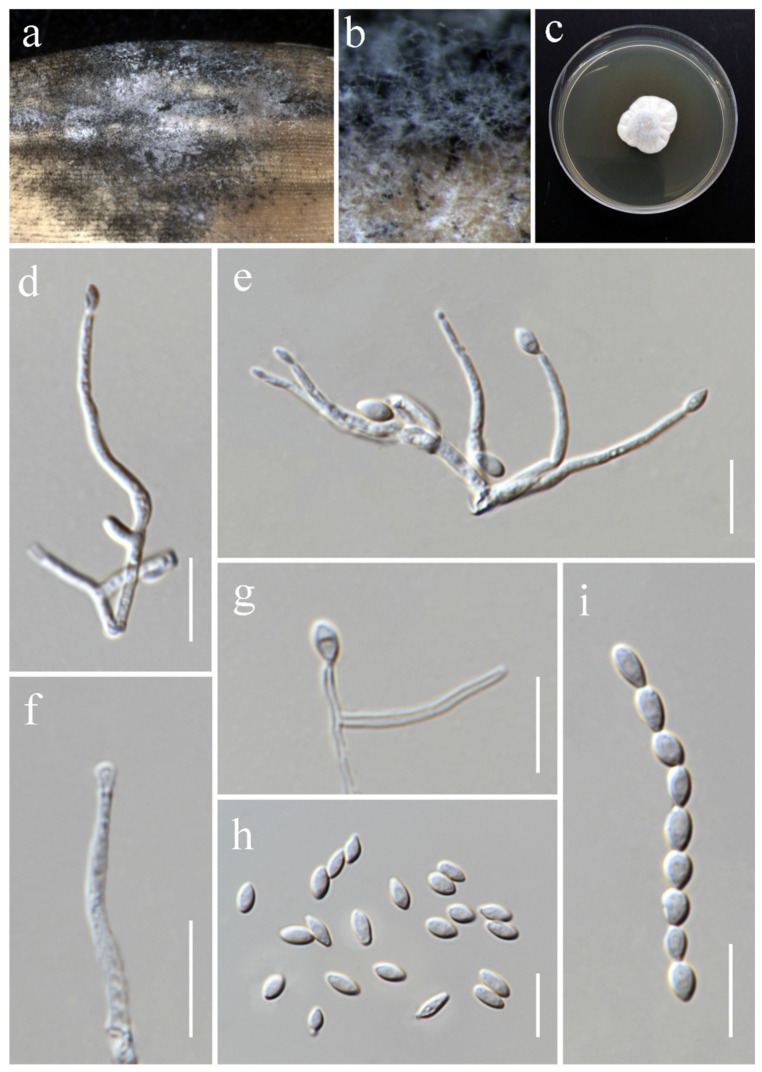
*Waltergamsia fusidioides* (MFLU 25-0044, new host and geographical record). (**a**,**b**) Appearance of mycelia on the panicle of *Oryza sativa*; (**c**) colony on PDA; (**d**–**g**) conidiophores; (**h**,**i**) conidia. Scale bars: (**d**–**i**) = 10 μm.

**Table 1 jof-11-00321-t001:** Primers and PCR conditions used in this study.

Locus	Primers	PCR Conditions	References
ITS	ITS5/ITS4	94 °C 3 min; 35 cycles of 94 °C 45 s, 56 °C 1 min, 72 °C 1 min; 72 °C 10 min	[60]
LSU	LROR/LR5	94 °C 3 min; 35 cycles of 94 °C 30 s, 55 °C 50 s, 72 °C 90 s; 72 °C 10 min	[59]
*tef*1-α	EF-1/EF-2	94 °C 90 s; 35 cycles of 94 °C 45 s, 55 °C 45 s, 72 °C 1 min; 72 °C 10 min	[61]
*rpb*2	RPB2–5F2/RPB2–7cR	94 °C 90 s; 40 cycles of 94 °C 30 s, 55 °C 30 s, 72 °C 2 min; 72 °C 10 min	[62]
*cmdA*	CAL-CL1/CAL-CL2A	94 °C 3 min; 35 cycles of 94 °C 30 s, 57 °C 30 s, 72 °C 1 min; 72 °C 10 min	[63]

## Data Availability

All sequence data are available in NCBI GenBank following the accession numbers in the manuscript.

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
