# Peer review of "Morpho-Molecular Characterization of Hypocrealean Fungi Isolated from Rice in Northern Thailand"

_jof, 2025, doi:10.3390/jof11040321_

Round 1
Reviewer 1 Report
The manuscript of Absalan with co-authors entitled “Morpho-Molecular Characterization of Hypocrealean Fungi Isolated from Rice in Northern Thailand” aimed the detailed study the biodiversity of this group of fungi. They revealed two new Fusarium species and described them. Authors reported four new host records and three new geographical records.
Some questions arise to improve the manuscript. 1) The number of samples, collection sites, frequency of findings. The most of fungi seem were found sporadically so their practical importance is low. 2) It would be nice to include information on the choice of phylogenetic analysis (1-8) for particular fungi.
Author Response
Comment 1: The number of samples, collection sites, frequency of findings. The most of fungi seem were found sporadically so their practical importance is low.
Response 1: We appreciate your valuable comments. Regarding the number of samples and collection sites, we were limited in our ability to expand sampling due to access restrictions. Collection from rice fields in Thailand requires the permission of the landowners or farmers. In several cases, the farmers were not present during our visits, and we could only collect specimens from fields where we had prior permission and were accompanied by a native Thai speaker to facilitate communication. As a result, our sampling was limited to specific sites where access was granted.
Concerning the statement that “most fungi seem to have been found sporadically, so their practical importance is low,” we would like to clarify that all specimens in this study were collected as saprobes, as noted in line 69: “The present study aims to investigate saprobic hypocrealean taxa associated with rice with emphasis on Fusarium in northern Thailand.” The primary aim of this study was taxonomic and phylogenetic characterization of fungi associated with rice ecosystems, particularly saprobic species, rather than an assessment of their practical or economic impact.
Comment 2: It would be nice to include information on the choice of phylogenetic analysis (1-8) for particular fungi.
Response 2:
Thank you for your insightful comment. In our study, we conducted Maximum Likelihood (ML) phylogenetic analyses using both RAxML and IQ-TREE. Based on a comparison of the resulting topologies and bootstrap support values, we found that the trees generated with IQ-TREE exhibited more consistent and well-resolved topologies with stronger bootstrap support. Therefore, we chose to present the IQ-TREE results in the manuscript.
Furthermore, we followed the methodology of several recent studies that conducted comprehensive phylogenetic analyses of taxa within the order Hypocreales. These references also supported the use of IQ-TREE as a more robust option for ML analyses within this group.
Reviewer 2 Report
I've carefully reviewed the manuscript "Morpho-Molecular Characterization of Hypocrealean Fungi Isolated from Rice in Northern Thailand" and can provide you with a comprehensive assessment to help with your publication decision.
Strengths
-
Comprehensive taxonomic study: The paper presents a thorough investigation of hypocrealean fungi associated with rice in Thailand, including the description of new species and new host/geographical records.
-
Methodological rigor: The authors employ a robust combination of morphological characterization and multi-locus phylogenetic analyses (ITS, LSU, tef1-α, rpb2, and cmdA), which is the current standard for fungal taxonomy.
-
Significant findings: The documentation of 14 species, including 2 new species (Fusarium chiangraiense and F. oryzigenum), 4 new host records, and 3 new geographical records represents a valuable contribution to mycology and plant pathology.
-
Detailed documentation: The descriptions of fungi are detailed and accompanied by high-quality photomicrographs that illustrate diagnostic features.
Areas for Improvement
-
Data availability: GenBank accession numbers for the sequences generated in this study are consistently missing. The manuscript repeatedly states "GenBank numbers –" without providing the actual numbers, which is essential for reproducibility.
-
Figures and phylogenetic trees: While mentioned in the text, many figures (including the phylogenetic trees) appear to be missing from the version I've reviewed. These are critical for validating the taxonomic claims.
-
Language and formatting: There are some formatting inconsistencies and minor grammatical errors throughout the manuscript:
-
Inconsistent spacing after periods
-
Some missing references in the Results section
-
Some paragraphs in the Discussion section could be better structured for clarity
-
-
Ecological context: While the paper focuses on taxonomy, more discussion on the ecological implications of these fungi (particularly the saprobic vs. pathogenic potential) would strengthen the manuscript's impact.
-
Phylogenetic analyses: The materials and methods section describes seven phylogenetic analyses, but in some cases, the relationship between these analyses and the taxonomic decisions could be more clearly explained.
Specific Concerns
-
The designation of new species should include more comparative discussion with closely related species, especially for F. chiangraiense and F. oryzigenum.
-
For some species (e.g., F. commune), the authors note they did not observe some morphological features described in the type specimen. These discrepancies need more thorough discussion.
-
The paper mentions the asexual morph of Ochronectria thailandica for the first time, but this significant observation deserves more detailed discussion.
Recommendation
Given the manuscript's strengths and the significance of the findings, I would recommend acceptance with minor revisions. The authors should:
-
Provide all missing GenBank accession numbers
-
Ensure all figures (especially phylogenetic trees) are properly included
-
Address the formatting and grammatical issues
-
Strengthen the comparative discussions for new species descriptions
-
Explain more clearly the ecological implications of their findings
The manuscript makes a valuable contribution to our understanding of fungal diversity associated with rice and provides important taxonomic updates in the Hypocreales. With the suggested revisions, it will be a worthy addition to the scientific literature in mycology and plant pathology.
I've carefully reviewed the manuscript "Morpho-Molecular Characterization of Hypocrealean Fungi Isolated from Rice in Northern Thailand" and can provide you with a comprehensive assessment to help with your publication decision.
Strengths
-
Comprehensive taxonomic study: The paper presents a thorough investigation of hypocrealean fungi associated with rice in Thailand, including the description of new species and new host/geographical records.
-
Methodological rigor: The authors employ a robust combination of morphological characterization and multi-locus phylogenetic analyses (ITS, LSU, tef1-α, rpb2, and cmdA), which is the current standard for fungal taxonomy.
-
Significant findings: The documentation of 14 species, including 2 new species (Fusarium chiangraiense and F. oryzigenum), 4 new host records, and 3 new geographical records represents a valuable contribution to mycology and plant pathology.
-
Detailed documentation: The descriptions of fungi are detailed and accompanied by high-quality photomicrographs that illustrate diagnostic features.
Areas for Improvement
-
Data availability: GenBank accession numbers for the sequences generated in this study are consistently missing. The manuscript repeatedly states "GenBank numbers –" without providing the actual numbers, which is essential for reproducibility.
-
Figures and phylogenetic trees: While mentioned in the text, many figures (including the phylogenetic trees) appear to be missing from the version I've reviewed. These are critical for validating the taxonomic claims.
-
Language and formatting: There are some formatting inconsistencies and minor grammatical errors throughout the manuscript:
-
Inconsistent spacing after periods
-
Some missing references in the Results section
-
Some paragraphs in the Discussion section could be better structured for clarity
-
-
Ecological context: While the paper focuses on taxonomy, more discussion on the ecological implications of these fungi (particularly the saprobic vs. pathogenic potential) would strengthen the manuscript's impact.
-
Phylogenetic analyses: The materials and methods section describes seven phylogenetic analyses, but in some cases, the relationship between these analyses and the taxonomic decisions could be more clearly explained.
Specific Concerns
-
The designation of new species should include more comparative discussion with closely related species, especially for F. chiangraiense and F. oryzigenum.
-
For some species (e.g., F. commune), the authors note they did not observe some morphological features described in the type specimen. These discrepancies need more thorough discussion.
-
The paper mentions the asexual morph of Ochronectria thailandica for the first time, but this significant observation deserves more detailed discussion.
Recommendation
Given the manuscript's strengths and the significance of the findings, I would recommend acceptance with minor revisions. The authors should:
-
Provide all missing GenBank accession numbers
-
Ensure all figures (especially phylogenetic trees) are properly included
-
Address the formatting and grammatical issues
-
Strengthen the comparative discussions for new species descriptions
-
Explain more clearly the ecological implications of their findings
The manuscript makes a valuable contribution to our understanding of fungal diversity associated with rice and provides important taxonomic updates in the Hypocreales. With the suggested revisions, it will be a worthy addition to the scientific literature in mycology and plant pathology.
Author Response
Comment 1: Provide all missing GenBank accession numbers.
Response 1: Thank you for pointing this out. All missing GenBank accession numbers have now been added to the revised manuscript.
Comment 2: Ensure all figures (especially phylogenetic trees) are properly included.
Response 2: Thank you for your comment. All figures, including the phylogenetic trees, were uploaded with the initial submission. It is possible that there was a technical issue during the upload or review process. We have rechecked the files and ensured that all figures are properly included in the revised version for your review.
Comment 3: Address the formatting and grammatical issues.
Response 3: We appreciate this remark. The manuscript has been reviewed for grammar and language issues, and the formatting has been adjusted to comply fully with the Journal of Fungi submission guidelines.
Comment 4: Strengthen the comparative discussions for new species descriptions (The designation of new species should include more comparative discussion with closely related species, especially for F. chiangraiense and F. oryzigenum).
Response 4: Thank you for this important observation. In Section 3.1, we provided molecular data and phylogenetic distinctions, while the comparative morphological characteristics and detailed notes for each new species, including F. chiangraiense and F. oryzigenum, are included in the Taxonomy part (Section 3.2). These notes contain direct comparisons with closely related species to support the designation of the new taxa.
Comment 5: Explain more clearly the ecological implications of their findings.
Response 5: Thank you for raising this point. As the focus of this study is primarily taxonomic, the ecological conclusions are inherently limited due to the scope of the data. We are cautious not to overstate ecological implications without direct evidence. However, we have included a general discussion based on previous studies in lines 681–697 to provide context regarding the potential ecological roles of these fungi within rice ecosystems.
Reviewer 3 Report
Sahar Absalan and colleagues present a taxonomic article on the identification and characterization of several fungi, isolated from rice in Thailand. Using morphological characteristics and phylogenetic analyses on different DNA sequences, several members from the family of Hypocreales were compared and discussed. In general, the article has a classical format used in several other publications of this group, it is comprehensibly written, the analyses appear to have been carefully conducted and it may contain interesting and important information for people working in this field. Below are a few comments that may help to improve the manuscript.
Given that this is an article focusing on morphological characteristics, all images depicting specific structures of the relevant fungal life cycle have to be very clear. This is not the case, as several images appear unfocused in several figures. Please provide new pics if necessary and improve the overall image quality.
Although I fully understand that a taxonomy based article is supposed to facilitate direct comparisons, still using the very same words and syntax in several paragraphs of the text is inappropriate. This is particularly evident in the entire chapter 3.1 (Phylogenetic analysis) and the Figure legends of Figures 1-7. Please revisit these parts and rephrase the relevant sentences.
Including coordinates of the geographical areas used for sampling would be informative.
Please also include the sequence identifiers (e.g. GenBank) for all these sequences in a relevant place of the manuscript.
please see major comments
Author Response
Comment 1: Given that this is an article focusing on morphological characteristics, all images depicting specific structures of the relevant fungal life cycle have to be very clear. This is not the case, as several images appear unfocused in several figures. Please provide new pics if necessary and improve the overall image quality.
Response 1: Thank you for this valuable feedback. All images included in the manuscript are prepared at 330 ppi, which is the highest resolution supported by the journal’s submission system. We acknowledge that photographing hyphomycetous fungi is technically more challenging than coelomycetous fungi due to the delicate and often minute structures involved. We have made every effort to capture and highlight the most relevant morphological features using the available equipment and facilities in our laboratory. Unfortunately, we no longer have access to the original specimens or cultures to retake the photographs. Nevertheless, we have carefully reviewed and, where possible, enhanced the clarity of the images to ensure they effectively support our descriptions.
Comment 2: Although I fully understand that a taxonomy-based article is supposed to facilitate direct comparisons, still using the very same words and syntax in several paragraphs of the text is inappropriate. This is particularly evident in the entire chapter 3.1 (Phylogenetic analysis) and the Figure legends of Figures 1–7. Please revisit these parts and rephrase the relevant sentences.
Response 2: Thank you for pointing this out. We agree that repetitive phrasing can reduce the clarity and readability of the text. We will revise Chapter 3.1 to improve the variation in language while maintaining accuracy and consistency in presenting the taxonomic comparisons.
Comment 3: Including coordinates of the geographical areas used for sampling would be informative.
Response 3: We appreciate this suggestion and will include the geographical coordinates of the sampling locations in the revised manuscript to provide more precise context for the study sites.
Comment 4: Please also include the sequence identifiers (e.g. GenBank) for all these sequences in a relevant place of the manuscript.
Response 4: Thank you for your suggestion. We will include the GenBank accession numbers for all sequences used in the study in the appropriate section of the revised manuscript.
Round 2
Reviewer 3 Report
I thank the authors for positively responding to the comments of the initial assessment. The revised manuscript has been improved. I have no further suggestions.
n.a.